# The YTHDF proteins ECT2 and ECT3 bind largely overlapping target sets and influence target mRNA abundance, not alternative polyadenylation

**Laura Arribas-Hernández[1]\*, Sarah Rennie[2], Michael Schon[3], Carlotta Porcelli[1], Balaji Enugutti[3], Robin Andersson[2], Michael D Nodine[3,4], Peter Brodersen[1]\***

[1]University of Copenhagen, Copenhagen Plant Science Center, Copenhagen N, Denmark; [2]University of Copenhagen, Department of Biology, Ole Maaløes Vej 5, Copenhagen, Denmark; [3]Gregor Mendel Institute (GMI), Austrian Academy of Sciences, Vienna Biocenter (VBC), Vienna, Austria; [4]Laboratory of Molecular Biology, Wageningen University, Wageningen, Netherlands

**Abstract** Gene regulation via *N6*-methyladenosine (m⁶A) in mRNA involves RNA-binding proteins that recognize m⁶A via a YT521-B homology (YTH) domain. The plant YTH domain proteins ECT2 and ECT3 act genetically redundantly in stimulating cell proliferation during organogenesis, but several fundamental questions regarding their mode of action remain unclear. Here, we use Hyper-TRIBE (targets of RNA-binding proteins identified by editing) to show that most ECT2 and ECT3 targets overlap, with only a few examples of preferential targeting by either of the two proteins. HyperTRIBE in different mutant backgrounds also provides direct views of redundant, ectopic, and specific target interactions of the two proteins. We also show that contrary to conclusions of previous reports, ECT2 does not accumulate in the nucleus. Accordingly, inactivation of *ECT2*, *ECT3*, and their surrogate *ECT4* does not change patterns of polyadenylation site choice in ECT2/3 target mRNAs, but does lead to lower steady-state accumulation of target mRNAs. In addition, mRNA and microRNA expression profiles show indications of stress response activation in *ect2/ect3/ect4* mutants, likely via indirect effects. Thus, previous suggestions of control of alternative polyadenylation by ECT2 are not supported by evidence, and ECT2 and ECT3 act largely redundantly to regulate target mRNA, including its abundance, in the cytoplasm.

**\*For correspondence:**
laura.arribas@bio.ku.dk (LA-H);
PBrodersen@bio.ku.dk (PB)

## Introduction

*N6*-methyladenosine (m⁶A) in mRNA is of fundamental importance in eukaryotic gene regulation (*Zhao et al., 2017*). It is deposited in the nucleus (*Salditt-Georgieff et al., 1976*; *Ke et al., 2017*; *Huang et al., 2019*) by the METTL3/METTL14-methyltransferase complex (MTA/MTB in plants) (*Bokar et al., 1997*; *Zhong et al., 2008*; *Liu et al., 2014*) and many functions of m⁶A involve RNA-binding proteins that recognize m⁶A in mRNA via a YT521-B homology (YTH) domain (*Stoilov et al., 2002*; *Dominissini et al., 2012*; *Wang et al., 2014*; *Zaccara et al., 2019*). The YTH-domain family is subdivided into two phylogenetic groups, YTHDF and YTHDC (*Patil et al., 2018*; *Balacco and Soller, 2019*), but the biochemistry used for m⁶A recognition is identical in both groups: an aromatic cage provides a hydrophobic environment for the *N6*-methyl group and stacking interactions with the adenine ring (*Li et al., 2014b*; *Luo and Tong, 2014*; *Theler et al., 2014*; *Wang et al., 2014*; *Xu et al., 2014*; *Zhu et al., 2014*). YTHDF proteins consist of an N-terminal intrinsically disordered region (IDR) followed by the YTH domain (*Patil et al., 2018*). Early reports seemed to indicate functional specialization of

vertebrate YTHDF proteins for either translational activation or mRNA decay (*Wang et al., 2014*; *Wang et al., 2015*; *Li et al., 2017*; *Shi et al., 2017*), whereas recent studies support functional redundancy among the three YTHDFs in mammals and zebrafish (*Kontur et al., 2020*; *Lasman et al., 2020*; *Zaccara and Jaffrey, 2020*), similar to the functional overlap described earlier for a subgroup of plant YTHDFs (*Arribas-Hernández et al., 2018*).

In plants, the YTHDF family is greatly expanded, with 11 members in *Arabidopsis*, referred to as EVOLUTIONARILY CONSERVED C-TERMINAL REGION1-11 (ECT1-11), compared to 3 in humans (*Li et al., 2014a*; *Scutenaire et al., 2018*). To date, only *ECT2*, *ECT3*, and *ECT4* have been studied using genetic approaches. These studies show that the three m⁶A readers have important functions in post-embryonic development, but appear to work largely redundantly, at least in formal genetic terms. This is because single knockouts of *ECT2* or *ECT3* produce only subtle phenotypes related to branching patterns of epidermal hairs and root growth directionality, while simultaneous knockout of *ECT2* and *ECT3* results in delayed organogenesis and defective morphology of leaves, roots, stems, flowers, and fruits; defects that are exacerbated by additional mutation of *ECT4* in most cases (*Arribas-Hernández et al., 2018*; *Arribas-Hernández et al., 2020*). It remains unclear, however, which mRNAs are targeted by ECT2/3, and what the effects of ECT2/3 binding to them may be (*Arribas-Hernández and Brodersen, 2020*). In particular, it is not clear whether the formal genetic redundancy between ECT2 and ECT3 is reflected in an overlapping target set, as would be expected for truly redundant action, or whether ECT2 and ECT3 might bind separate targets in wild type plants, but are able to replace each other in the artificial situation created by gene knockouts. The fact that knockouts of *ECT2* and *ECT3* have opposite effects on root growth directionality (*Arribas-Hernández et al., 2020*) indicates that at least some level of functional specialization exists between them, despite the obvious genetic redundancy observed in the control of organogenesis. Thus, it is an open question of fundamental importance for understanding plant m⁶A-YTHDF axes whether, and to what degree, mRNA targets of ECT2 and ECT3 overlap.

ECT2 has previously been suggested to act in the nucleus to influence alternative polyadenylation of targets (*Wei et al., 2018*). This model implies that plant ECT2 would act fundamentally differently from metazoan YTHDF proteins that are thought to be exclusively cytoplasmic and act to control mRNA fate via accelerated mRNA decay, or translational status (*Patil et al., 2018*; *Zaccara et al., 2019*; *Kan et al., 2021*; *Worpenberg et al., 2021*), perhaps in some cases by influencing the ability of other RNA binding proteins to associate with specific mRNAs (*Worpenberg et al., 2021*). The evidence for nuclear localization of ECT2 is not unequivocal, however, because the ECT2 signal presumed to be nuclear has not been examined relative to a nuclear envelope marker. In contrast, all studies examining the subcellular localization of ECT2 (and ECT3 and ECT4) have clearly established their presence in the cytoplasm (*Arribas-Hernández et al., 2018*; *Scutenaire et al., 2018*; *Wei et al., 2018*; *Arribas-Hernández et al., 2020*). In addition, the model of ECT2-mediated gene regulation via alternative polyadenylation has not been tested by direct experimentation.

Here, we use the proximity-labeling method HyperTRIBE (targets of RNA binding proteins identified by editing) (*McMahon et al., 2016*; *Xu et al., 2018*) to identify mRNA targets of ECT3. HyperTRIBE uses fusion of an RNA-binding protein to a hyperactive mutant (E488Q) of the catalytic domain of the *Drosophila melanogaster* adenosine deaminase acting on RNA (*Dm*ADARcd) to obtain an A-G mutation profile specifically in mRNAs bound by the RNA-binding protein of interest in vivo. We combine the comparative analysis of this data set with the target identification of ECT2 by HyperTRIBE and iCLIP (individual nucleotide resolution crosslinking and immunoprecipitation) (*König et al., 2010*) reported in the accompanying paper (*Arribas-Hernández et al., 2021*), a series of transcriptomic analyses in *ect2/ect3/ect4* triple knockout mutants, and super-resolution microscopy of ECT2 localization, to establish three fundamental properties of mRNA regulation by ECT2 and ECT3. (1) Most targets are shared between ECT2 and ECT3, and the two proteins act genuinely redundantly in vivo to bind to and regulate many targets, in agreement with their similar expression patterns and genetically redundant functions (*Arribas-Hernández et al., 2018*; *Arribas-Hernández et al., 2020*). (2) ECT2/3/4 do not appreciably influence alternative polyadenylation of target mRNAs, consistent with the absence of ECT2-mCherry from the nucleoplasm. (3) In ECT2-expressing cell populations, the abundance of the majority of ECT2/3-target mRNAs is reduced upon loss of ECT2/3/4 activity.

## Results

### Identification of ECT3 target mRNAs using HyperTRIBE

To identify target mRNAs of ECT3 transcriptome-wide, we chose HyperTRIBE, because of our demonstration in the accompanying paper (*Arribas-Hernández et al., 2021*) that it efficiently identifies ECT2 targets with little expression bias. We therefore proceeded in exactly the same way as described for ECT2, in this case using transgenic lines expressing *AtECT3pro:AtECT3-FLAG-DmADAR^E488Q^cd-AtECT3ter* (henceforth '*ECT3-FLAG-ADAR*') in *ect3-1* single mutants, and of *FLAG-ADAR* under the control of the *ECT3* promoter (henceforth simply '*FLAG-ADAR*') in wild type background as negative control (*Figure 1A*). Complementation of the developmental phenotype of triple *ect2-1/ect3-1/ect4-2* (*te234*) mutants (*Arribas-Hernández et al., 2018*) by expression of *ECT3-FLAG-ADAR* at comparable levels was also verified (*Figure 1A*, *Figure 1—figure supplement 1A*). Five lines of each type were used for mRNA-seq of dissected shoot and root apices, and the data was analyzed to identify differentially edited sites (*Figure 1B*, *Figure 1—figure supplements 1 and 2*). Despite the lower expression of *ECT3* compared to *ECT2* (*Arribas-Hernández et al., 2018*) and, consequently, generally lower editing proportions in *ECT3-FLAG-ADAR* lines compared to *ECT2-FLAG-ADAR* lines (*Arribas-Hernández et al., 2021*; *Figure 1C*), the implementation of the HyperTRIBE**R** pipeline (*Rennie et al., 2021*) to call significant editing sites successfully identified 2448 targets in aerial tissues, and 3493 in roots (ECT3 HT-targets) (*Figure 1B*, *Supplementary file 1*). As seen for ECT2 (*Arribas-Hernández et al., 2021*), the ECT3 target genes in shoot and root apices largely overlapped, and the editing proportions of individual editing sites showed a strong correlation (*Figure 1D and E*). Accordingly, most aerial- or root-specific targets could be explained by differences in expression between tissues (*Figure 1F*). The identification of strongly overlapping target sets in roots and shoots is expected from the similar roles of ECT3 in promoting growth and cell division in the two tissues (*Arribas-Hernández et al., 2020*) and, therefore, constitutes an argument for robustness of ECT3 target identification by the HyperTRIBE method.

### ECT2 and ECT3 bind to overlapping sets of targets

We next analyzed the degree to which ECT2 and ECT3 HT-targets overlap. The data sets are directly comparable, as growth conditions, tissue dissection, RNA extraction, library construction, and sequencing depth for target identification of ECT3 by HyperTRIBE were identical to those used for ECT2 (*Arribas-Hernández et al., 2021*; *Figure 2—figure supplement 1*). Remarkably, more than 94 % of ECT3 HT-targets overlapped with the larger group of ECT2 HT-targets in both aerial and root tissues, and there was a clear correlation between the editing proportions of the common editing sites, albeit with higher editing by ECT2-FLAG-ADAR overall (*Figure 2A and B*). Indeed, the pattern of editing sites resulting from fusion of ADAR to ECT2 or ECT3 was similar for many targets, with a few more sites typically detected in the ECT2-HT data set (e.g., *ATPQ*, *Figure 2C*, *left panel*). Nevertheless, we also noticed examples with preferential targeting by ECT2-FLAG-ADAR (e.g., *TUA4*, *Figure 2C*, *middle panel*) or, interestingly, by the less abundant ECT3-FLAG-ADAR (e.g., *UBQ6*, *Figure 2C*, *right panel*), perhaps hinting to molecular explanations for the recently described non-redundant roles of ECT2 and ECT3 in determining root growth directionality (*Arribas-Hernández et al., 2020*). Overall, however, the overwhelming overlap between ECT2 and ECT3 HT-targets in both tissues suggests that binding to the same mRNA targets underlies their genetically redundant functions in leaf and root formation (*Arribas-Hernández et al., 2018*; *Arribas-Hernández et al., 2020*). We also observed that *ECT2* and *ECT3* mRNAs contain m⁶A sites in seedlings according to published data sets (*Shen et al., 2016*; *Parker et al., 2020*), and that their protein products target their own and each other's transcripts (*Figure 2—figure supplement 2*), indicating that autoregulatory feedback may contribute to control their expression.

### HyperTRIBE provides direct views of redundant target mRNA Interactions with ECT2 and ECT3

Although the demonstration that ECT2 and ECT3 bind to strongly overlapping target sets is consistent with largely redundant in vivo function, it does not constitute a direct proof. For example, the proteins may bind to the same targets, but in different cells such they act de facto non-redundantly. We reasoned that HyperTRIBE might provide a means to observe directly whether ECT2 and ECT3

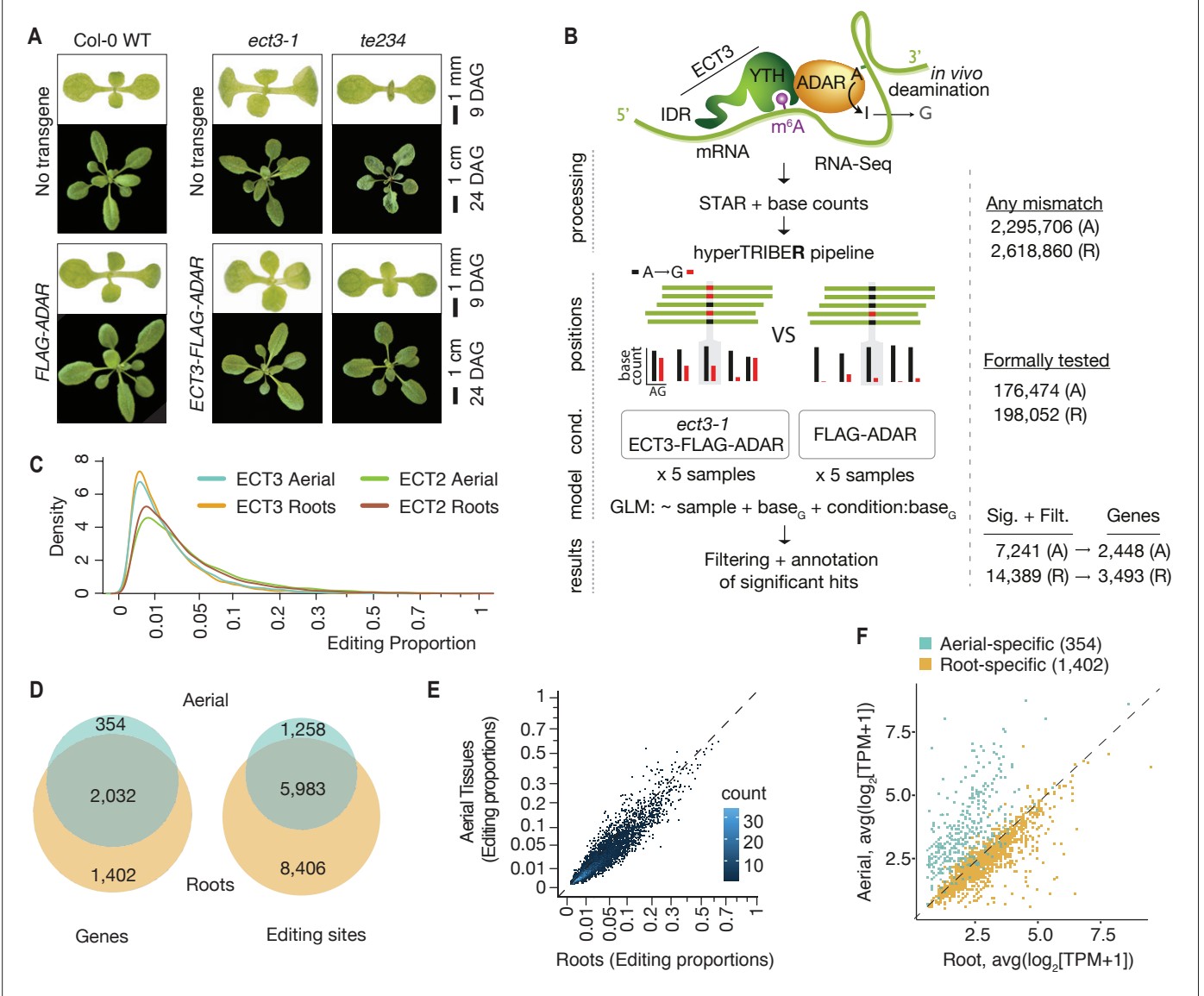

**Figure 1.** Identification of ECT3 targets using HyperTRIBE. (**A**) Phenotypes of wild type, *ect3-1*, and *te234* (*ect2-1/ect3-1/ect4-2*) mutants with (lower panels) or without (upper panels) *ECT3pro:ECT3-FLAG-DmADAR^{E488Q}cd-ECT3ter* (*ECT3-FLAG-ADAR*) or *ECT3pro:FLAG-DmADAR^{E488Q}cd-ECT3ter* (*FLAG-ADAR*) transgenes, at 9 or 24 days after germination (DAG). (**B**) Experimental design for ECT3-HyperTRIBE (ECT3-HT) target identification. After quantifying nucleotide base counts from mapped RNA-seq libraries of *ect3-1/ECT3-FLAG-ADAR* and *FLAG-ADAR* lines, all positions with mismatches were passed into the HyperTRIBER pipeline (*Rennie et al., 2021*) to call significant editing sites. Identified sites were further filtered to remove SNPs and retain only A-to-G mismatches. The number of sites in either aerial (A, dissected apices) or root (R, root tips) tissues at each stage is indicated. (**C**) Density of editing proportions for significant editing sites in aerial tissues and roots of *ect3-1/ECT3-FLAG-ADAR* and *ect2-1/ECT2-FLAG-ADAR* (*Arribas-Hernández et al., 2021*) lines. (**D**) Overlap between ECT3-HT target genes and editing sites in roots and aerial tissues, out of the set of genes commonly expressed in both tissues. (**E**) Scatterplot of the editing proportions of significant editing sites in ECT3-HT for aerial versus root tissues. (**F**) Scatterplot showing the expression in aerial and root tissues (mean log$_2$ (TPM +1) over the five ECT3-HT control samples) of the genes that are identified as targets only in aerial tissues or only in roots. GLM, generalized linear model.

The online version of this article includes the following figure supplement(s) for figure 1:

**Figure supplement 1.** Identification of ECT3 targets using HyperTRIBE (extended data).

**Figure supplement 2.** Characteristics of ECT3-HyperTRIBE editing sites relative to target expression levels.

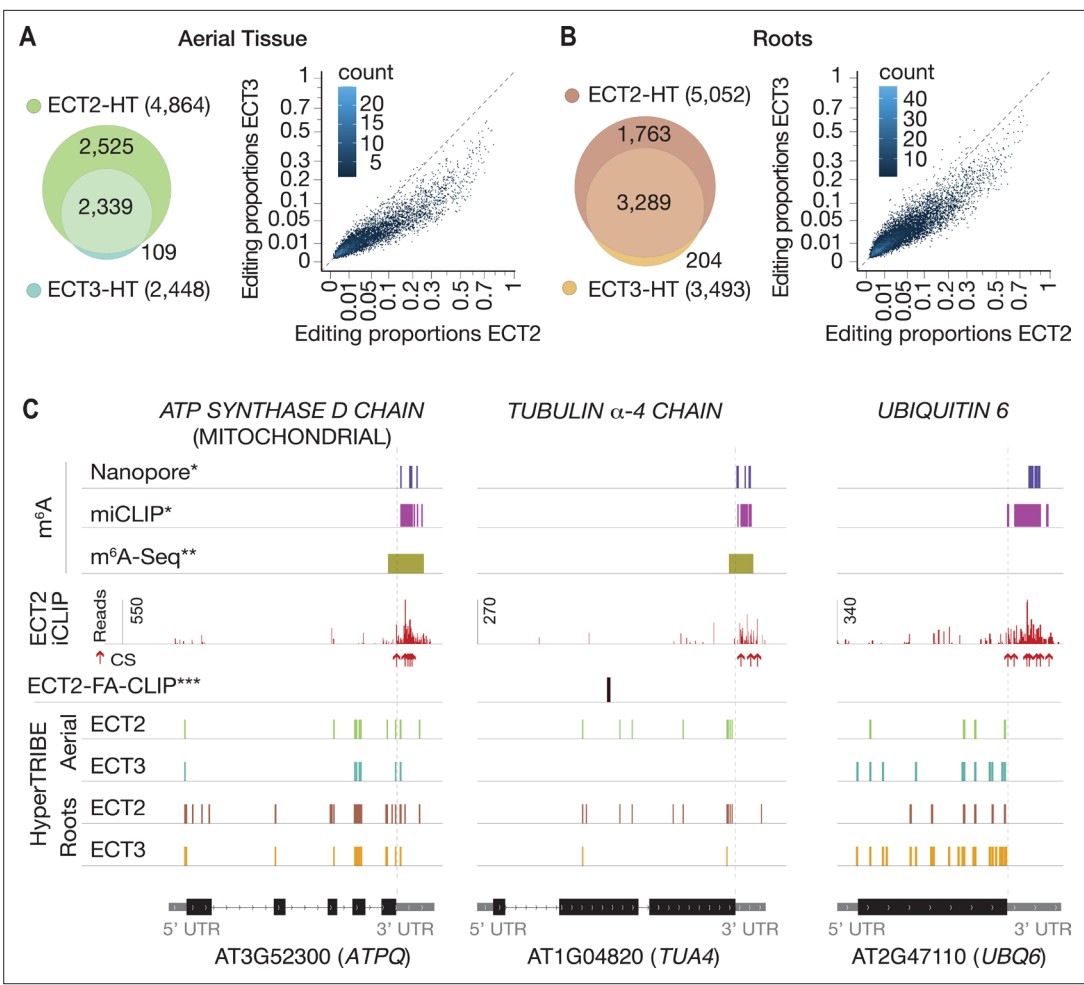

**Figure 2.** ECT3 targets are fewer and largely contained within ECT2 targets. (**A, B**) Left panels: overlap of ECT2-HT and ECT3-HT targets, for the set of commonly expressed genes, in aerial (**A**) and root (**B**) tissues. Right panels: Scatter plots showing the editing proportions of editing sites between ECT2-HT and ECT3-HT, for all significant positions common to both sets, separately for aerial (**A**) and root (**B**) tissues. (**C**) Examples of common ECT2 and ECT3 targets showing the distribution of ECT2/3-HT editing sites in either aerial or root tissues along the transcript. The distribution of ECT2-iCLIP reads and crosslink sites (CS) (*Arribas-Hernández et al., 2021*), FA-CLIP peaks***, and m$^6$A sites*, ** is also shown. * *Parker et al., 2020*; ** *Shen et al., 2016*; *** *Wei et al., 2018*.

The online version of this article includes the following figure supplement(s) for figure 2:

**Figure supplement 1.** Sequencing depth of ECT2 and ECT3 HyperTRIBE RNA-seq data.

**Figure supplement 2.** ECT2 and ECT3 target each other and themselves.

act specifically or redundantly on shared targets, and whether one ECT protein acquires non-natural targets upon knockout of the other by comparison of editing proportions measured with ADAR fusions expressed in single versus triple mutant backgrounds. The single mutant background (e.g., *ECT2-FLAG-ADAR* in *ect2-1*) would mimic the wild type setting, while the triple mutant background (e.g., *ECT2-FLAG-ADAR* in *te234*) would probe target interactions in the absence of redundant or competing proteins, but still in plants with wild type growth rates (*Figure 1A; Arribas-Hernández et al., 2021*). Redundant target interactions would be expected to result in generally higher editing proportions of the same target sites in triple mutant than in single mutant backgrounds, especially for the least expressed protein, ECT3 (*Figure 3—figure supplement 1*). Conversely, specific interactions would cause one of two possible signatures. (i) In the case of cell type-specific interactions, no change in editing proportions between single and triple mutant backgrounds should be detectable. (ii) In the case of specific interactions within the same cells in wild type, but non-natural targeting in the absence of other ECT proteins, acquisition of non-natural targets in triple mutant backgrounds is expected.

We observed widespread increases in editing proportions for ECT3 targets upon removal of ECT2/4, while such increases occurred only sporadically for ECT2 targets (*Figure 3A–C*, *Supplementary file 2*), supporting the idea of largely redundant target interactions. Furthermore, although more sites showed increased editing by ECT3-FLAG-ADAR in aerial tissues than in roots in the absence of ECT2/4 (*Figure 3B*), the net increase of editing proportions was higher in roots for both ECT2 and ECT3 (*Figure 3C*), consistent with a more dominant role of ECT2 over ECT3 in aerial tissues compared to roots (*Figure 2A and B*). Importantly, the higher editing proportions in the triple mutant background cannot be trivially explained by higher expression of the transgene in these lines, as the average expression was comparable or slightly lower (*Figure 3—figure supplement 1*). These observations directly support genuinely redundant interactions of ECT2 and ECT3 with the majority of their mRNA targets in vivo.

## Small sets of specific ECT2/3 targets acquire unnatural ECT interactions in knockout backgrounds

Although the tendency of ECT2/3 to show redundant target mRNA interactions was widespread, we also looked for examples of specific interactions in the HyperTRIBE data in single and triple mutant backgrounds. A priori, we considered targets to be ECT2-specific if they were edited by ECT2-FLAG-ADAR, but not ECT3-FLAG-ADAR, in single mutants (strictly specific), or became edited by ECT3-FLAG-ADAR only in the triple mutant. The definition of ECT3-specific targets followed analogous criteria. However, because ECT2 expression is much higher than ECT3 expression (*Figure 3—figure supplement 1*), ECT2-specific targets identified in this way may simply be below the detection limit of the less expressed ECT3-FLAG-ADAR transgene. Hence, arguments for existence of bona fide specific targets must take detectability by ECT3-FLAG-ADAR into account. Consistent with the expectation from the different ECT2/ECT3 dosage, much larger numbers of strictly ECT2-specific transcripts were identified compared to ECT3: 2414 ECT2-specific and 93 ECT3-specific targets were identified in aerial tissues, while in roots, 1738 were ECT2-specific and 197 were ECT3-specific (*Figure 3D and E*, *Figure 3—figure supplement 2*). In addition, small sets of specific target mRNAs became targets of the other ECT protein upon knockout of its genuine interacting protein (110 and 24 for ECT2-specific targets in aerial and root tissues, respectively, and 2 for ECT3-specific targets in roots) (*Figure 3D and E*, *Figure 3—figure supplement 2*). These sets constitute outstanding candidates for ECT2/3-specific mRNA targets. Interestingly, mRNAs of four tubulin subunits (*TUBA1*, *TUBB1*, *TUBB5*, and *TUBB9*) were among the many ECT2-specific targets in roots. This finding may be biologically relevant, because *ect2* and *ect3* single mutants exhibit distinct root slanting phenotypes (*Arribas-Hernández et al., 2020*), and misregulation of tubulin subunits causes root slanting defects (*Smyth, 2016*). Curiously, a few transcripts (21 in aerial tissues that include four photosynthesis-related genes [*LHCB1.2*, *LHB1B2*, *LHB1B1*, and AT3G63540], and 9 in roots) were edited by either ECT2 or ECT3 only in the triple mutant background (*Figure 3D and E*, *Figure 3—figure supplement 2*). Because *ECT4* is also knocked out in *te234*, these sets define putative ECT4-specific targets. In summary, our comparative analyses of ECT2/3 HyperTRIBE data obtained in single and triple mutant backgrounds indicate that redundant target interaction is pervasive, but they also identify small target sets with properties consistent with preferential interaction with only one ECT protein.

## ECT2/3 targets tend to be co-expressed in proliferating cells and are enriched in functions related to basic metabolism and protein synthesis

We next combined the ECT3-HT target set described here with the ECT2 iCLIP and ECT2-HT data (*Arribas-Hernández et al., 2021*) to define three gene sets of particular interest for functional analysis of ECT2/3: **The permissive target set** (6528 genes) defined as genes with either ECT2 HT, ECT3 HT, or ECT2 iCLIP support, the **stringent target set** (1992 genes) defined as all ECT2 or ECT3 HT-targets that are also in the ECT2 iCLIP target set, and **the non-target set** (13,504 genes) defined as all expressed genes not contained in the permissive target set (*Figure 4A*, *Figure 4—figure supplement 1*, *Supplementary file 3*). As an initial check of consistency of the target sets with the biological context in which ECT2 and ECT3 function, we used single-cell transcriptome analysis of *Arabidopsis* roots (*Denyer et al., 2019*; *Ma et al., 2020*) to analyze the overlap of ECT2/3 expression with the enrichment of markers for different cell types in the permissive target set (roots only, *Figure 4—figure supplement 1*). This analysis showed reassuring congruence between predominant expression

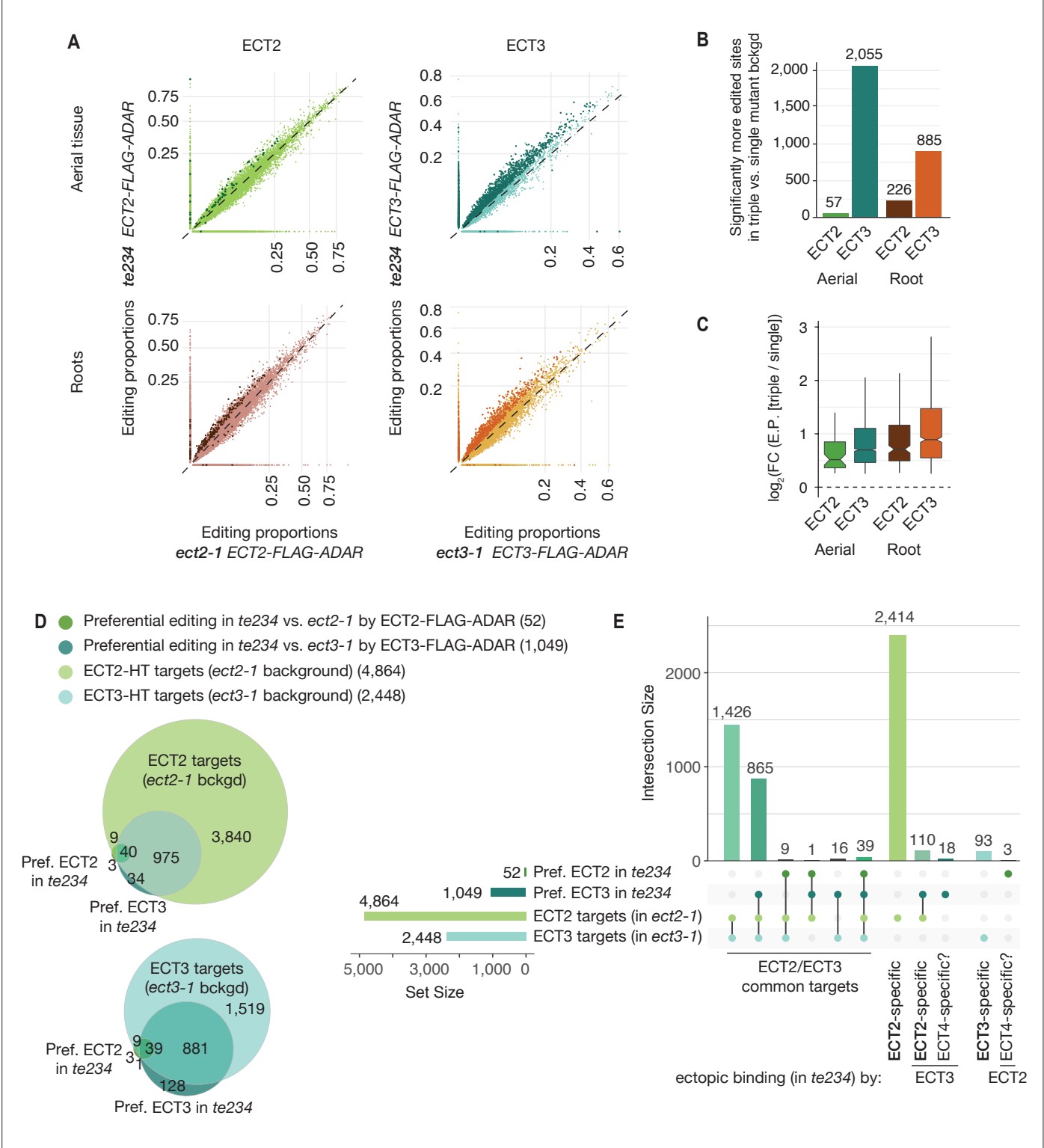

**Figure 3.** Redundancy between ECT2 and ECT3. (**A**) Scatterplots comparing the editing proportions of ECT2- and ECT3-FLAG-ADAR observed in triple versus single mutant backgrounds in aerial and root tissues. They include all positions significantly edited with respect to FLAG-ADAR controls (p-value<0.01, log$_2$(FC) >1) in either background, with dots on the axes reflecting positions not significantly edited in one of the two backgrounds. Dots in darker shades indicate positions more highly edited in one background compared to the other (p-value<0.1, log$_2$(FC) >0.25 or log$_2$(FC) <−0.25). (**B**) Barplots showing the number of positions significantly more edited in triple versus single mutant background for each tissue and ECT protein. Positions significantly less edited in the triple mutant background were fewer than 12 in all cases. (**C**) Boxplots showing fold changes in editing proportions

*Figure 3 continued on next page*

*Figure 3 continued*

between the triple and single mutant backgrounds for the two ECT proteins and tissues studied. (**D, E**) Venn diagrams (**D**) and Upset plot (**E**) showing the overlap between the ECT2-HT (*Arribas-Hernández et al., 2021*) and ECT3-HT target sets (in single mutant backgrounds) with the groups of genes with more highly edited positions in the triple mutant background in aerial tissues (the equivalent for roots is shown in *Figure 3—figure supplement 2*).

The online version of this article includes the following figure supplement(s) for figure 3:

**Figure supplement 1.** Expression levels (TPM) of the FLAG-ADAR-containing transgenes in all HyperTRIBE lines.

**Figure supplement 2.** Overlap between ECT2/3-HT targets in single and triple mutant backgrounds in roots.

---

of ECT2/3 in meristem clusters and marker enrichment for these same clusters among targets (*Figure 4B*, *Figure 4—figure supplement 2*). We also analyzed the permissive target sets for groups of functionally related genes, and found that ECT2/3 targets are enriched in housekeeping genes, many related to basic metabolism and protein synthesis (*Figure 4C*). This result is in agreement with the functional enrichment analysis of m⁶A-containing transcripts in mature leaves (*Anderson et al., 2018*) and of genes differentially expressed in rosettes of mutants partially depleted of m⁶A (*Bodi et al., 2012*; *Anderson et al., 2018*). Together with the recurrent function of ECT2/3/4 in organogenesis (*Arribas-Hernández et al., 2020*), these results suggest that regulation of metabolism and

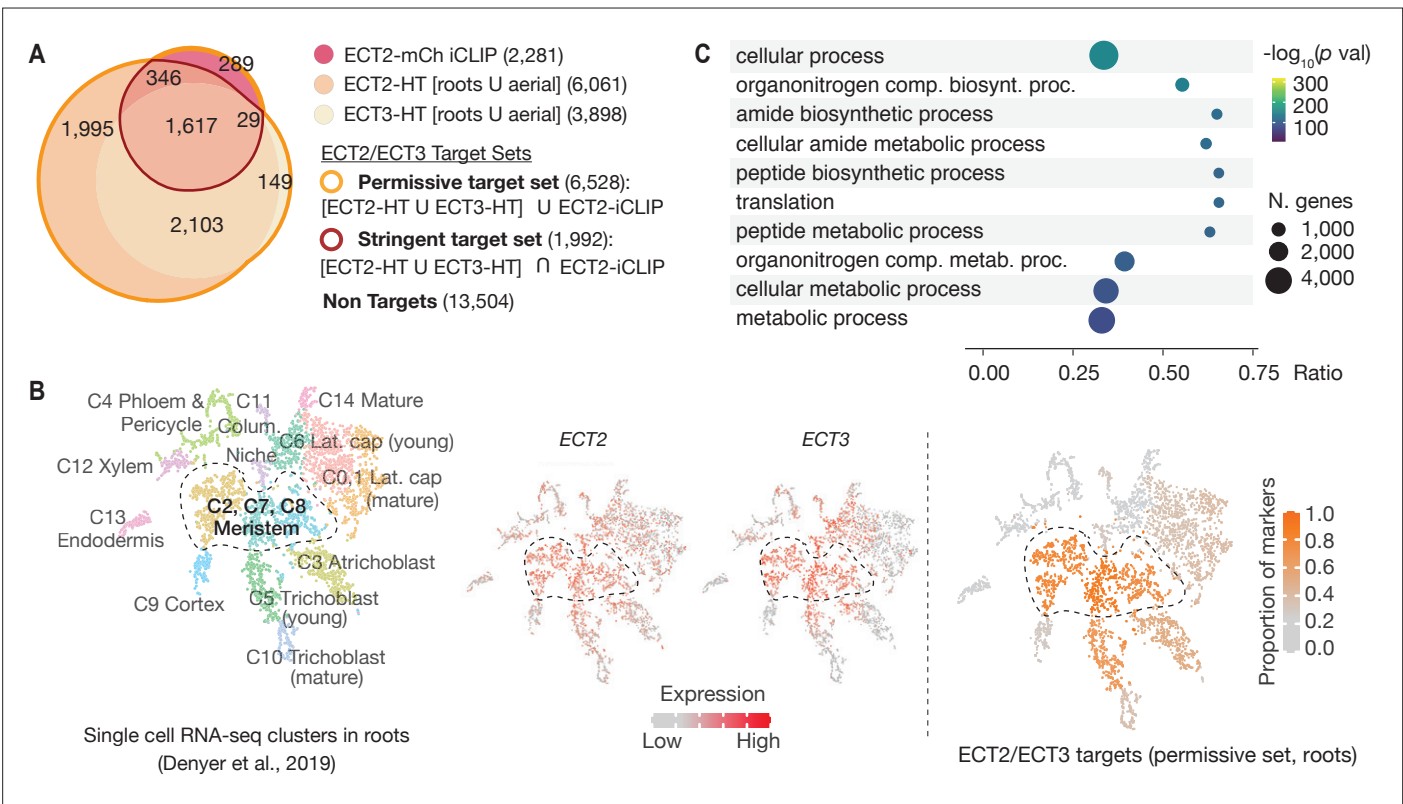

**Figure 4.** ECT2 and ECT3 targets are co-expressed with *ECT2* and *ECT3* in proliferating cells and enriched in biosynthetic processes. (**A**) Overlap between ECT2-iCLIP target genes with ECT2-HT and ECT3-HT target gene sets. Regions outlined in bold orange and red indicate the defined permissive and stringent ECT2/3 target sets in whole seedlings, respectively (aerial and root-specific target sets are shown in *Figure 4—figure supplement 1*). Non-targets are all expressed genes (with detectable transcript levels in the corresponding HyperTRIBE RNA-seq data sets) that are not in the permissive target set. (**B**) Left: t-SNE plot for scRNA-seq data in roots from *Denyer et al., 2019*, with cells colored according to their cell-type cluster definitions (see *Figure 4—figure supplement 2* for details). Center: *ECT2* and *ECT3* single-cell expression levels overlayed on to the t-SNE plot (*Ma et al., 2020*). Right: t-SNE plot with cell-type clusters shaded according to the proportion of marker genes from *Denyer et al., 2019* that are targets of ECT2 or ECT3 in roots. Dashed enclosed region indicates clusters that contain meristematic cells. (**C**) The 10 most significantly enriched GO terms among ECT2/3 targets (permissive set). GO, gene ontology.

The online version of this article includes the following figure supplement(s) for figure 4:

**Figure supplement 1.** ECT2 and ECT3 target sets in aerial and root tissues (separately).

**Figure supplement 2.** ECT2/3 targets are co-expressed with *ECT2/3* in highly dividing root cells.

---

growth is the main biological function of the m⁶A-ECT2/3/4 regulatory axis. These initial analyses also provide well-defined common ECT2/3 target sets for further functional studies.

## Recovery of ECT2-expressing cell populations with and without ECT2/ECT3/ECT4 activity

*ECT2*, *ECT3*, and *ECT4* expression is largely restricted to rapidly dividing cells of organ primordia (*Arribas-Hernández et al., 2018*; *Arribas-Hernández et al., 2020*), and since many ECT2/3 targets are broadly expressed housekeeping genes (*Figure 4C*), cell populations expressing ECT2–4 need to be isolated prior to transcriptome analyses to avoid confounding effects from cells that do not express these m⁶A readers. We therefore used the fact that *ect2-1/ECT2-mCherry* exhibits root growth rates similar to wild type while *te234/ECT2$^{W464A}$-mCherry* exhibits clearly reduced root growth rates nearly identical to *te234* triple knockouts (*Arribas-Hernández et al., 2020*), and applied fluorescence-associated cell sorting to select mCherry-expressing cell populations from root protoplasts of three independent transgenic lines for each of these two genetic backgrounds (*Figure 5A*). Because wild type and mutant fluorescent proteins have the same expression level, pattern, and intracellular local-ization (*Figure 5B and C*), this procedure yielded comparable ECT2-expressing cell populations (*Figure 5D*, *Figure 5—figure supplement 1*) with (*ECT2-mCherry/ect2-1* henceforth 'wild type') or without (*ECT2$^{W464A}$-mCherry/ect2-1/ect3-1/ect4-2,* henceforth 'mutant') ECT2/3/4 function. We therefore isolated mRNA and constructed Smart-Seq2 libraries for comparison of poly(A) sites (PASs) and abundance of ECT2/3 targets and non-targets in *ECT2*-expressing cells from plants of the two different genetic backgrounds. Compared to standard mRNA-seq, Smart-seq2 recovers more reads with untemplated As (beginning of poly(A) tails) in addition to gene-specific sequence and can, there-fore, be used for PAS mapping. We note that the selection of ECT2-expressing cells from the root meristem of wild type and mutant lines also circumvents the trouble of preparing comparable samples from intact tissues of plants at different developmental stages despite having the same age.

## ECT2/3/4 do not play a direct role in alternative polyadenylation of targets

We first addressed the conjecture on a nuclear role of ECT2 in PAS selection (*Wei et al., 2018*). In plants, PASs are not sharply defined but rather spread along localized regions and can be grouped into PAS clusters (PACs) for analysis (*Wu et al., 2011*; *Sherstnev et al., 2012*). Using a modification of the nanoPARE analysis pipeline (*Schon et al., 2018*) to map PASs from reads with ≥9 untemplated As, we identified a total of 14,667 PACs belonging to 12,662 genes after filtering possible false positives (see Materials and methods; *Figure 5—figure supplement 2A,B*, *Supplementary file 4*). We found no tendency for ECT2/3 target mRNAs to have more PACs than non-targeted genes (*Figure 5—figure supplement 2C*), suggesting that differential PAC location in ECT2/3 targets between mutant and wild type is not prevalent. Nevertheless, we specifically tested whether PASs could be affected by the loss of ECT2/3/4 function in two different ways: either a shift of the dominant PAC to an alter-native PAC altogether, or a shift in the most common PAS within clusters. Sorting the 206 genes for which the dominant PAC differed between wild type and mutant samples (18.5 % of the 1114 genes with more than one PAC) into the ECT2/3 target groups in roots (*Figure 4—figure supplement 1B*, *Supplementary file 4*) showed that both the permissive and stringent targets were significantly less likely than non-targets to have a different dominant PAC upon loss of ECT2/3/4 function (p=0.013 and p=1.21e−5 for strictly permissive and stringent targets, respectively; Fisher's exact test) (*Figure 5E*, *Figure 5—figure supplement 2C,D*). This significant difference may be an effect of the higher expres-sion of targets compared to non-targets (*Arribas-Hernández et al., 2021*), as accuracy of PAS detec-tion increases with transcript abundance (see *Figure 5—figure supplement 2E* for details). The result indicates that the alternative polyadenylation observed upon loss of ECT2/3/4 function is not prev-alent among ECT2/3 targets. Finally, we examined changes to the local distribution of PASs within clusters. We defined the most common PAS as the single position in all overlapping PACs with the most reads, and determined the distance between such dominant PASs in wild type and mutant samples. Comparison of the distances revealed that the most common PAS does not change by more than 10 bp in the majority of genes, and is not more likely to be different in ECT2/3 targets than in non-targets (*Figure 5F*). In fact, the most common PAS is more likely to be unchanged in targets than in non-targets (*Figure 5G*) (p=0.028 and p=2.2e−16 for strictly permissive and stringent targets,

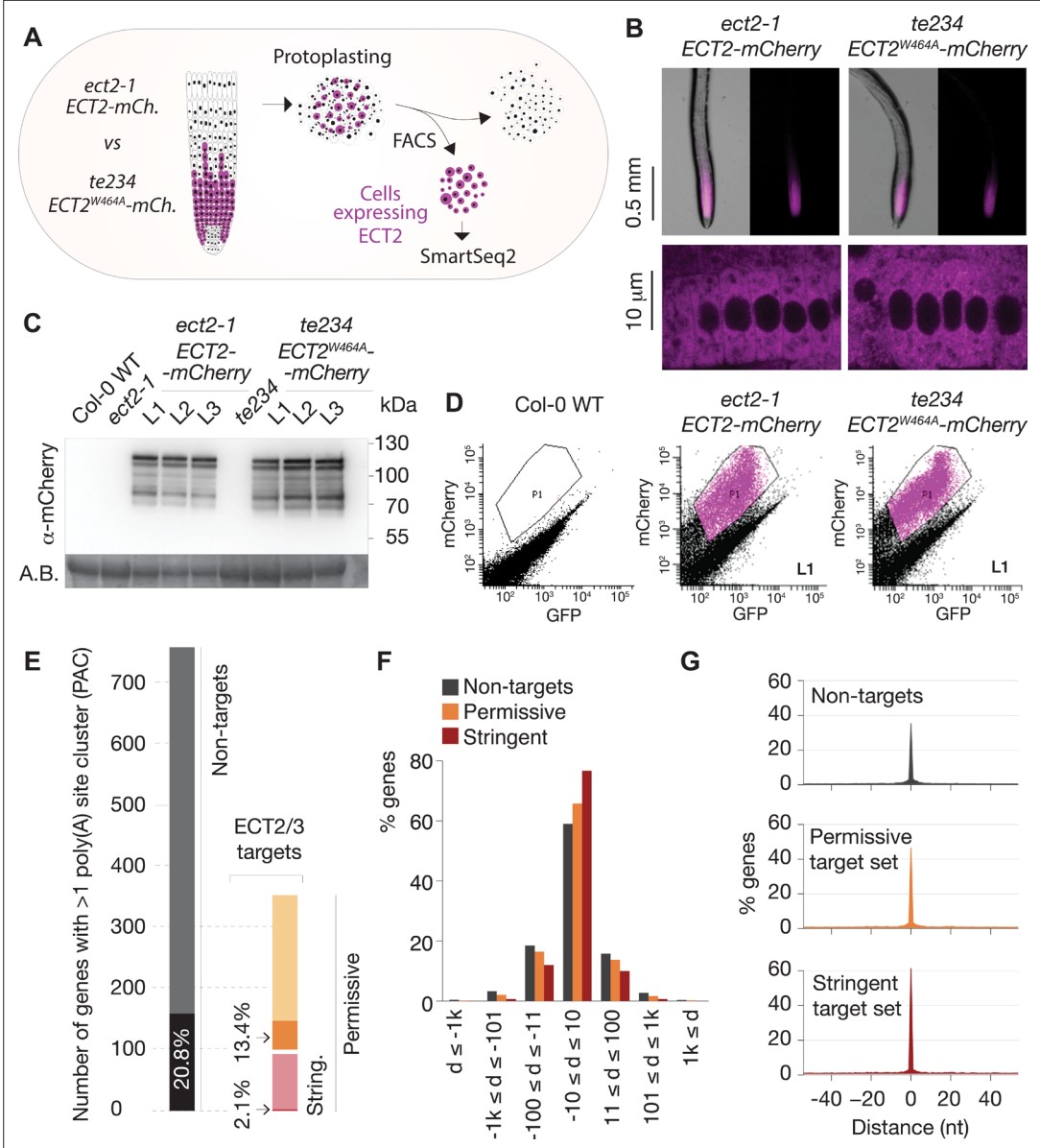

**Figure 5.** Poly(A) sites in ECT2/3 targets do not change upon loss of ECT2/3/4 function. (**A**) Experimental design. The experiment was performed once, using three biological replicates (independent lines) per group (genotype). (**B**) Expression pattern of ECT2-mCherry in root tips of *ect2-1 ECT2-mCherry* and *te234 ECT2^{W464A}-mCherry* genotypes by fluorescence microscopy. (**C**) Protein blot showing expression levels of ECT2-mCherry in the 3+3 lines of *ect2-1 ECT2-mCherry* and *te234 ECT2^{W464A}-mCherry* used as biological replicates for FACS selection of ECT2-expressing cells. Amido black (A.B.) is used as loading control. (**D**) Fluorescence profile (mCherry vs. GFP fluorescence) of root cells (protoplasts) from the transgenic lines in (**C**). The complete set of lines/samples is shown in *Figure 5—figure supplement 1*. Non-transgenic Col-0 WT is shown as control for background autofluorescence. Cells with a fluorescence profile within the outlined areas were selected for RNA extraction, Smart-seq2 library construction, and sequencing. (**E**) Genes with more than one poly(A) site cluster (PAC) in the different target/non-target sets. Dark shades are genes in which the dominant PAC in *te234 ECT2^{W464A}-mCherry* samples differs from the one in *ect2-1 ECT2-mCherry*. (**F, G**) Distribution of distances (d [nt]) of the most common poly(A) site between *te234 ECT2^{W464A}-mCherry* and *ect2-1 ECT2-mCherry* samples for all genes where the most common poly(A) site could be determined in both genotypes (6648 non-targets, 4072 permissive targets, and 1486 stringent targets). Negative values are upstream (5') and positive values are downstream (3') relative to the gene orientation. (**F**) Distances are binned by ±10, ±100, ±1000, and >1000 bp. (**G**) Distances are plotted by nucleotide in a ±40 bp window.

*Figure 5 continued on next page*

*Figure 5 continued*

The online version of this article includes the following source data and figure supplement(s) for figure 5:

**Source data 1.** Original (uncropped) membrane from *Figure 5C*.

**Figure supplement 1.** FACS-sorting of root protoplasts expressing ECT2-mCherry.

**Figure supplement 2.** Poly(A) sites do not change in ECT2/3 targets upon loss of ECT2/3/4 function (extended data).

respectively; Fisher's exact test). Taken together, these analyses show that neither the usage of alternative PACs nor the dominant PASs within clusters have any tendency to change in ECT2/3 targets upon loss of ECT2/3/4 function.

## ECT2-mCherry does not localize to the nucleoplasmic side of the nuclear envelope

To further investigate whether ECT2 may have any nuclear functions, we revisited the evidence for localization of ECT2 in the nucleoplasm, which is based on confocal fluorescence microscopy of ECT2-GFP or YFP-ECT2 in DAPI-stained root cells of stable *Arabidopsis* lines (*Scutenaire et al., 2018*; *Wei et al., 2018*). Because (i) the localization of ECT2-mCherry in living root cells of our lines has a general sharp boundary with what we interpreted to be the nucleus (*Figure 5B*; *Arribas-Hernández et al., 2018*) and does not overlap with nucleoplasmic MTA-TFP (*Arribas-Hernández et al., 2020*), (ii) paraformaldehyde fixation routinely used to permeate DAPI inside plant tissues (used by Scutenaire et al. and not specified by Wei et al.) can introduce artifacts in the localization of fluorescent proteins (*Man-Wah et al., 2015*), and (iii) the RNA-binding properties of DAPI could yield signal from the RNA-rich rough endoplasmic reticulum surrounding the nucleus (*Tanious et al., 1992*), we decided to examine the localization of ECT2-mCherry relative to the nuclear envelope in living cells. We therefore crossed lines expressing functional ECT2-mCherry (*Arribas-Hernández et al., 2020*) with plants expressing the outer nuclear envelope and nuclear pore complex-associated protein WIP1 fused to GFP (*Xu et al., 2007*). Confocal fluorescence microscopy of intact roots showed that the sharp boundaries of the ECT2-mCherry expression domain were delimited by the GFP-WIP1 signal from the nuclear envelope (*Figure 6A*, *Figure 6—figure supplement 1A*). Importantly, the occasional points at which the ECT2-mCherry signal seemed to fuzzily spill into the nucleus (white arrows in *Figure 6A*, *Figure 6—figure supplement 1A*) overlapped with equally blurry GFP-WIP1 signal, probably due to lack of perpendicularity between the nuclear envelope and the optical section in these areas. In such cases, the cytoplasm, nucleus and nuclear envelope may be contained in the same region of the optical section and thus appear to be overlapping (*Figure 6A, bottom panel*). To verify this interpretation, we inspected our plants with the super-resolution confocal Airyscan detector (*Huff, 2015*) and, as expected, we did not observe ECT2-mCherry signal inside the GFP-WIP1-delimited nuclei in any instances (*Figure 6B*, *Figure 6—figure supplement 1B,C*). Based on these analyses, we conclude that ECT2 resides in the cytoplasm and its presence in the nucleus, if any, may be too transient to be detected by fluorescence microscopy. These results agree with the lack of evidence for a function of ECT2/3/4 in choice of PAS, and strongly suggest that the molecular basis for the importance of ECT2/3/4 should be sought in cytoplasmic properties of their mRNA targets.

## ECT2/3 targets tend to show reduced abundance upon loss of ECT2/3/4

We next assessed the effect of loss of ECT2/3/4 function on target mRNA abundance, using the Smart-seq2 data from FACS-sorted root protoplasts described above. Principal component analysis showed that the three repeats of 'wild type' (*ect2-1/ECT2-mCherry*) were well separated from the three repeats of 'mutant' (*te234/ECT2$^{W464A}$-mCherry*) along the first principal component (*Figure 7—figure supplement 1A*), indicating that the differential gene expression analysis was meaningful. We focused on stringent, permissive, and non-ECT2/3 targets in roots (*Figure 4—figure supplement 1B*, *Supplementary file 3*), and visualized their differential expression by scatter, volcano, and box plots (*Figure 7A–C*, *Supplementary file 5*). These approaches showed that stringent targets have a clear tendency toward downregulation upon loss of ECT2/3/4 function. This trend is maintained, but is less pronounced in permissive targets, and is reversed in non-targets (*Figure 7A–C*). Indeed, of

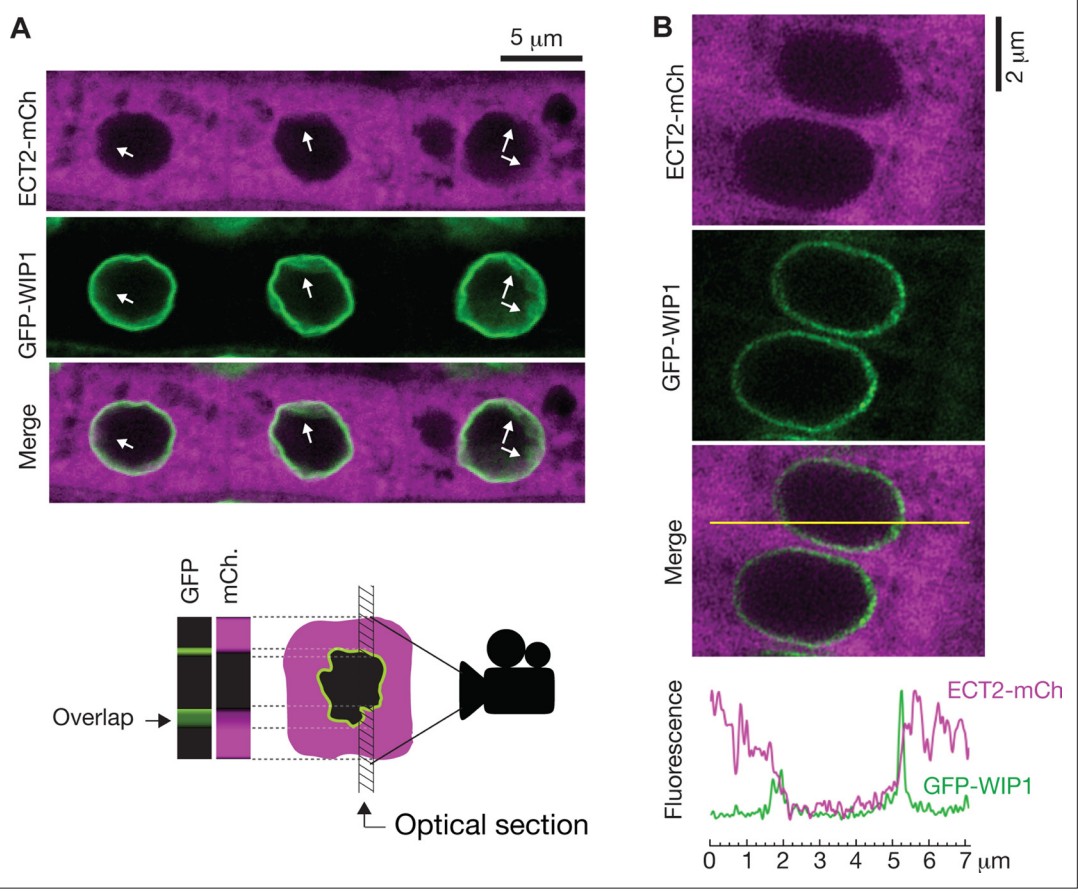

**Figure 6.** ECT2 is not in the nucleus. (**A**) Standard confocal microscopy of root cells co-expressing *ECT2-mCherry* and *GFP-WIP1*. White arrows indicate areas in which apparent spills of ECT2-mCherry signal into the nucleus overlap with blurry GFP signal from the nuclear envelope, a sign of not-perpendicularity between the envelope and the optical plane as exemplified on the cartoon at the bottom. (**B**) Airyscan super-resolution confocal microscopy of root cells as in (**A**). The image is cropped from a larger picture shown in *Figure 6—figure supplement 1*. mCherry and GFP fluorescence signals along the yellow line show absence of ECT2-mCherry inside the limits of the GFP-labeled nuclear envelope.

The online version of this article includes the following figure supplement(s) for figure 6:

**Figure supplement 1.** Super-resolution confocal microscopy of cells co-expressing ECT2-mCherry and the nuclear envelope marker GFP-WIP1.

the significantly differentially expressed stringent ECT2/3 targets, nearly all were downregulated in the mutant, while the majority of differentially expressed non-targets were upregulated compared to wild type (*Figure 7D*). Furthermore, ECT2/3 targets accounted for more than half of all significantly downregulated genes, but only about 15 % of upregulated genes (*Figure 7E*). In contrast, highly upregulated genes tended to be non-targets (*Figure 7B*, *right panel*).

## Functional groups of differentially expressed genes

To test if these differentially regulated gene sets represented subsets of functionally related genes within target and non-target groups, we analyzed their potential enrichment of gene ontology terms. This analysis revealed that downregulated ECT2/3 targets were particularly enriched in genes related to ribosome biogenesis and translation (*Figure 7F*), while upregulated non-targets were enriched in 'abiotic stress responses' with molecular function 'transcription factor' (*Figure 7G*). Because cell wall digestion required for protoplast isolation is a cellular stress, we tested the trivial possibility that loss of ECT2/3/4 function renders cells more susceptible to stress, and that such potential hyper-susceptibility underlies the observed differences of gene expression in ECT2-expressing root protoplasts. To this end, we isolated RNA from intact root apices of 4 -day-old plants of Col-0 wild

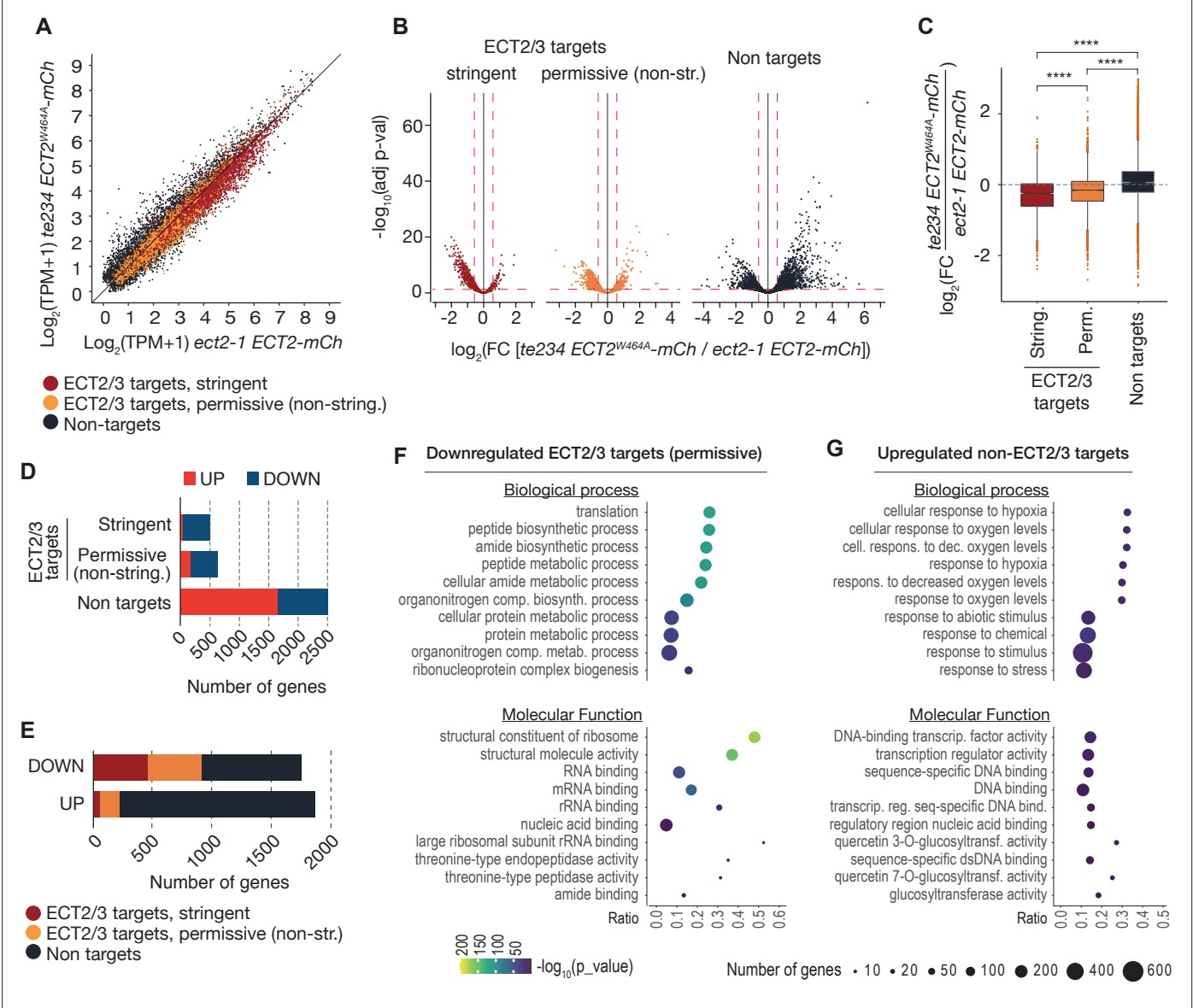

**Figure 7.** ECT2/3 targets are generally less abundant in cells without ECT2/3/4 function. (**A**) Scatterplot of TPM expression values in Smart-seq2 libraries of root protoplasts expressing ECT2-mCherry in *te234/ECT2^W464A^-mCherry* versus *ect2-1/ECT2-mCherry* samples. (**B**) Volcano plots reveal genes differentially expressed between the genotypes described in (**A**). (**C**) Boxplots of log$_2$ fold change expression values between *te234/ECT2^W464A^-mCherry* and *ect2-1/ECT2-mCherry* samples. (**D, E**) Bar plots showing the amount of significantly upregulated and downregulated genes in ECT2/3 targets and non-targets. (**F, G**) List with the 10 most significantly enriched GO terms among significantly downregulated ECT2/3 targets (permissive set) (**F**), or upregulated non-targets (**G**) upon loss of ECT2/3/4 function. GO, gene ontology.

The online version of this article includes the following figure supplement(s) for figure 7:

**Figure supplement 1.** Differential expression analysis using intact root tips of *ect2/ect3/ect4* knockout plants recapitulates the results obtained with sorted protoplasts.

type and *te234* mutants, and performed mRNA-seq analysis. These results recapitulated the trends of downregulation of stringent ECT2/3 targets and upregulation of stress-responsive non-targets, albeit with less pronounced differences than observed in the selected ECT2-expressing cell populations as expected (*Figure 7—figure supplement 1*). We also noticed that several stress-inducing and growth-restricting NF-YA-class transcription factors, all repressed by miR169defg (*Song et al., 2019*), were upregulated in root tips (*Figure 8A*), and used small RNA-seq to test if activation of the stress response was visible in the miRNA expression profile. Indeed, the miR169defg family was

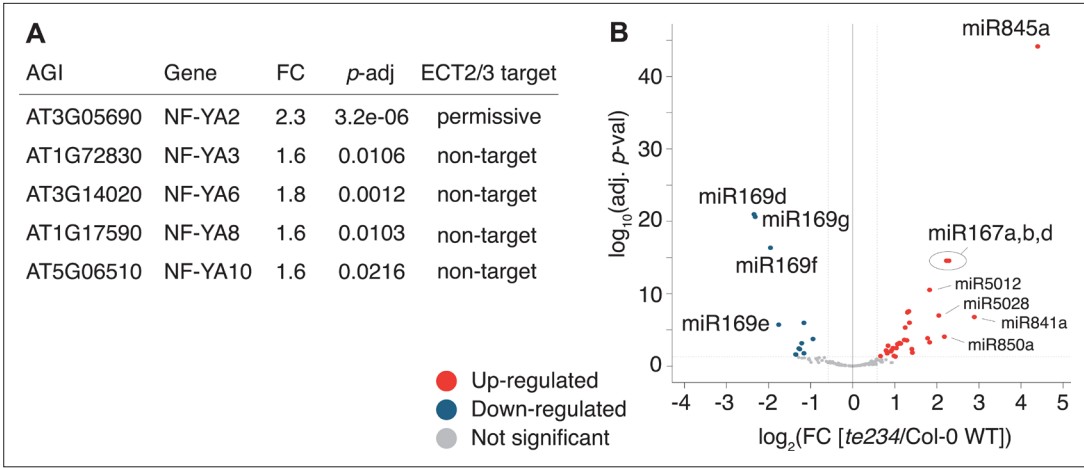

**Figure 8.** miRNA profile in root tips of *ect2/ect3/ect4* knockout plants. (**A**) Differential expression analysis of *te234* versus Col-0 WT in 4-day-old root tips shows that expression of several stress-related NF-YA transcription factors (miR169defg targets) is induced in *te234* plants. FC, Fold Change [*te234*/WT]. (**B**) Volcano plot showing miRNAs differentially expressed in *te234* versus Col-0 WT in 4-day-old root tips.

specifically repressed in *te234* mutants, and the miR167 family, targeting growth-promoting auxin-response factors, was clearly upregulated (*Figure 8B*). In addition, the LTR-retrotransposon-targeting miR845a (*Borges et al., 2018*) was strongly upregulated. Thus, the stress response detected in *te234* mutants comprises coherent changes of miRNA and transcription factor expression. Because it has been recently reported that processing of some miRNAs is affected by MTA-dependent methylation of their precursors in *Arabidopsis* (*Bhat et al., 2020*), we queried our iCLIP and HyperTRIBE data for evidence of direct binding of ECT2/3 to pri-miRNAs, possibly explaining the differential abundance of the miR169, miR167, and miR845 families in *te234* mutants. However, we could not find evidence for ECT2/3 binding to these or any other pri-miRNA in our data, consistent with the cytoplasmic localization of ECT2 and the predominantly, though not exclusively, nuclear functions of pri-miRNA (*Fang and Spector, 2007*; *Lauressergues et al., 2015*). Thus, the differential abundance of mature miRNAs in roots of *te234* seedlings compared to wild type is likely to be caused by the same indirect effects that lead to induction of stress-related mRNAs.

Altogether, the analysis of data from intact root tips confirms that the observed patterns of differential gene expression in selected protoplasts are genuine and biologically meaningful, and that the selection of ECT2-expressing cells ensures the most accurate description of differential gene expression resulting from loss of ECT2/3/4 function. We note that while the differential gene expression analysis suggests that ECT2/3/4 formally act to increase abundance of their mRNA targets, it does not allow conclusions to be drawn on how such activation is brought about: a direct stabilizing effect of ECT2/3/4 binding to their targets is consistent with the observed results, but indirect effects via transcriptional repression cannot be excluded, especially given the presence of stress-related transcription factors in the set of upregulated non-targets.

## Discussion

Our identification and comparative analyses of mRNA targets of ECT2 (*Arribas-Hernández et al., 2021*) and ECT3, and the study of target behavior in terms of abundance and use of alternative polyadenylation in cell populations devoid of ECT2/3/4 activity, allow us to draw conclusions on (i) the redundancy, and (ii) the subcellular localization and functional impact of these YTHDF proteins in *Arabidopsis*.

### Redundancy within the YTHDF protein family in plants

Combining the overlapping expression patterns of ECT2 and ECT3, their formal genetic redundancy in growth promotion (*Arribas-Hernández et al., 2018*; *Arribas-Hernández et al., 2020*), their overlapping target sets, and the signatures of redundant target interaction derived from HyperTRIBE in

single and triple *ect* mutant backgrounds, we conclude that many target mRNAs can bind to either ECT2 or ECT3 with similar consequences. Thus, ECT2 and ECT3 can exhibit redundant function sensu stricto, not just the ability to replace function in the absence of the other protein. However, this does not mean that the *ECT2* and *ECT3* genes fulfill identical function under all conditions. For example, a recent study of the yeast stress-related transcription factors MSN2 and MSN4 shows that distributing fully redundant biochemical function over two genes expressed in the same cell can be beneficial, because strong induction by stress can be combined with steady low-noise expression under favorable conditions (*Chapal et al., 2019*); a combination difficult, if not impossible, to obtain with control of expression by a single promoter.

Looking at the complete ECT family beyond ECT2 and ECT3, our previous studies may offer additional insight into redundancy with ECT4. Because of the enhancement of developmental phenotypes of *ect2/ect3* plants by mutation of *ECT4*, and the overlapping expression pattern of the three proteins (*Arribas-Hernández et al., 2018*; *Arribas-Hernández et al., 2020*), ECT4 may act redundantly with ECT2/3 on the same targets. The reduced impact of absence of ECT4 compared to that of ECT2/3 may simply be due to lower expression levels, but could also involve lower binding affinity, reduced activity, or a combination of those properties. However, such conclusions on potentially redundant function cannot be drawn for any of the eight additional members of the YTHDF family in *Arabidopsis*. Indeed, RNA-seq data suggests that at least some YTHDFs are expressed in different tissues. For example, *ECT10* mRNA is highly abundant in pollen from which the rest of the family is absent (*Arribas-Hernández et al., 2018*). Although it is possible that specific expression patterns or induction by environmental cues are the main reasons for the expansion of the YTHDF family in plants (*Li et al., 2014a*; *Scutenaire et al., 2018*), functional specialization that may include different target-binding specificity (*Arribas-Hernández et al., 2021*) and effector function conferred by the different IDRs is also possible. For those ECTs with overlapping expression patterns, competition for targets leading to different molecular outcomes is an interesting possibility. Further genetic and biochemical studies on additional ECTs are necessary to understand redundant and specific functions of YTHDF proteins, a question that is currently under debate also in animal systems (*Kontur et al., 2020*; *Lasman et al., 2020*; *Zaccara and Jaffrey, 2020*).

## Subcellular localization and functional impact on mRNA targets of plant YTHDF proteins

Our data show that ECT2 is not nuclear, and that ECT2/3/4 do not appreciably affect alternative polyadenylation in their direct mRNA targets. It is important to note that this conclusion on ECT2/3/4 does not extend to m⁶A altogether. Polyadenylation and transcription termination are clearly influenced by m⁶A in plants, as shown by studies of mutants in core *N6*-adenosine methyltransferase components VIR and FIP37. In *vir-1* mutants, a tendency to use proximal alternative PASs in m⁶A-targets is prominent (*Parker et al., 2020*), and in *fip37* mutants, several cases of defective transcription termination causing the production of chimeric transcripts were noted (*Pontier et al., 2019*). In the latter case, m⁶A recognition by the YTHDC-containing nuclear subunit of the Cleavage and Polyadenylation Specificity Factor CPSF30 was demonstrated to be required for regular transcription termination (*Pontier et al., 2019*), and subsequent studies of CPSF30, including of mutants specifically defective in m⁶A-binding, also indicated its role in mediating m⁶A-dependent alternative polyadenylation (*Hou et al., 2021*; *Song et al., 2021*). Thus, m⁶A does affect 3′-end formation of methylated pre-mRNAs in plants, but this process involves the nuclear CPSF30 which contains a YTHDC domain only in plants, rather than the cytoplasmic YTHDF protein ECT2.

The immediate implication of the conclusions that ECT2 does not act in the nucleus and does not influence alternative polyadenylation is that the conceptual framework for m⁶A-YTHDF action established mostly through studies in mammalian cell culture does extend to plants: m⁶A is installed by a nuclear methyltransferase complex (*Zhong et al., 2008*; *Shen et al., 2016*; *Ružička et al., 2017*), probably coupled to RNA Polymerase II transcription (*Bhat et al., 2020*), and YTHDF-mediated regulation of transcripts carrying the m⁶A mark takes place in the cytoplasm. Then, what is the molecular effect of ECT2/ECT3-binding to target mRNAs? Clearly, lack of ECT2/3/4 results in decreased abundance of direct targets. This result is in line with the lower accumulation of m⁶A-containing transcripts observed in plants partially depleted of m⁶A (*Shen et al., 2016*; *Anderson et al., 2018*; *Parker et al., 2020*) and in the *ect2-1* knockout line (*Wei et al., 2018*) compared to wild type plants. However,

differential expression analysis of ubiquitously expressed targets performed with RNA from entire plants is not easy to interpret when the regulatory process subject to study is located only in well-defined cell populations of meristematic tissues (*Zhong et al., 2008*; *Arribas-Hernández et al., 2018*; *Arribas-Hernández et al., 2020*). The problem is even more acute when the tissue composition of wild type and mutant individuals to be compared is different due to the developmental delay of the m⁶A-deficient lines. Finally, some of the previous studies of differential gene expression also revealed a number of m⁶A-containing transcripts with increased abundance in m⁶A-deficient mutants (*Shen et al., 2016*; *Anderson et al., 2018*), of which some were proposed to be of importance for the observed phenotype (*Shen et al., 2016*). Thus, the studies on differential gene expression in m⁶A-deficient mutants compared to wild type published thus far do not allow clear conclusions on the consequence of loss of m⁶A (or a reader protein) for target mRNA accumulation to be drawn (*Arribas-Hernández and Brodersen, 2020*). In contrast, our experimental setup, using a tissue with comparable cell-type composition in wild type and mutant plants and extracting RNA only from the cells in which target regulation by ECT2/3/4 takes place, establishes that targets generally have decreased abundance upon loss of ECT2/3/4.

The present study does not, however, elucidate the mechanisms involved in target regulation, because it does not directly measure target mRNA synthesis and degradation rates. It is possible that the reduced target mRNA accumulation in *ect2/3/4* mutant cells is exclusively a direct consequence of ECT2/3/4 function at the post-transcriptional level. For example, ECT2/3/4 could cause mRNA stabilization by protection from endonucleolysis as previously suggested (*Anderson et al., 2018*). It is also possible that ECT2/3/4 act primarily to repress target mRNA translation, as suggested by the recent demonstration that although *Drosophila* Ythdf translationally represses the *futsch* target mRNA, *futsch* mRNA abundance is reduced in *ythdf* mutants (*Worpenberg et al., 2021*). We cannot at present exclude, however, that more indirect effects also play a role, perhaps related to transcriptional repression of ECT2/3/4 targets via stress responses activated upon loss of ECT2/3/4 function. We anticipate that clear answers to this question must await development of tools for conditional inactivation of ECT2/3 function, such that consequences for mRNA target stability and translatability can be studied immediately after loss of ECT2/3 binding. Finally, we note that the constitutive stress response activation is consistent with the stunted phenotype of *mta* and *te234* mutants (*Bodi et al., 2012*; *Arribas-Hernández et al., 2018*; *Arribas-Hernández et al., 2020*), and that the earliest report of differentially expressed genes in an m⁶A-deficient organism, performed in plant leaves, also showed enrichment of stress-responsive genes, interpreted by the authors as possibly, '*the consequence of a perceived stress due to reduced methylase activity*' (*Bodi et al., 2012*). Thus, disentangling the direct effects of m⁶A-ECT2/3/4 on growth via mRNA target regulation from possible indirect effects arising from stress response activation in knockout mutants will be of major importance in future studies.

## Materials and methods

**Key resources table**

| Reagent type (species) or resource | Designation | Source or reference | Identifiers | Additional information |
|---|---|---|---|---|
| Gene (*Arabidopsis thaliana*) | ECT2 | TAIR10 | AT3G13460 | *EVOLUTIONARILY CONSERVED C-TERMINAL REGION 2* |
| Gene (*Arabidopsis thaliana*) | ECT3 | TAIR10 | AT5G61020 | *EVOLUTIONARILY CONSERVED C-TERMINAL REGION 3* |
| Gene (*Arabidopsis thaliana*) | ECT4 | TAIR10 | AT1G55500 | *EVOLUTIONARILY CONSERVED C-TERMINAL REGION 4* |
| Gene (*Arabidopsis thaliana*) | WIP1 | TAIR10 | AT4G26455 | *WPP DOMAIN INTERACTING PROTEIN 1* |
| Gene (*Drosophila melanogaster*) | ADAR Isoform N | Genebank, FlyBase, NCBI | CG12598 NM_001297862 | Adenosine deaminase acting on RNA |

*Continued on next page*

*Continued*

| Reagent type (species) or resource | Designation | Source or reference | Identifiers | Additional information |
|---|---|---|---|---|
| Genetic reagent (*A. thaliana*) | SALK_002225 C (*ect2-1*) | NASC | N657472, N2110120 | |
| Genetic reagent (*A. thaliana*) | SALK_077502 (*ect3-1*) | NASC | N577502, N2110123 | |
| Genetic reagent (*A. thaliana*) | *te234 (ect2-1/ect3-1/ect4-2)* | *Arribas-Hernández et al., 2018* | N2110132 | Donated to NASC and ABRC |
| Genetic reagent (*A. thaliana*) | *ECT3pro:FLAG-DmADAR$^{E488Q}$cd-ECT3ter* | This paper (see Methods) | | Seed requests to pbrodersen@bio.ku.dk |
| Genetic reagent (*A. thaliana*) | *ect3-1/ECT3pro:ECT3-FLAG-DmADAR$^{E488Q}$cd-ECT3ter* | This paper (see Methods) | | Seed requests to pbrodersen@bio.ku.dk |
| Genetic reagent (*A. thaliana*) | *te234/ECT3pro:ECT3-FLAG-DmADAR$^{E488Q}$cd-ECT3ter* | This paper (see Methods) | | Seed requests to pbrodersen@bio.ku.dk |
| Genetic reagent (*A. thaliana*) | *ect2-1/ECT2pro:ECT2-FLAG-DmADAR$^{E488Q}$cd-ECT2ter* | *Arribas-Hernández et al., 2021* | | |
| Genetic reagent (*A. thaliana*) | *te234/ECT2pro:ECT2-FLAG-DmADAR$^{E488Q}$cd-ECT2ter* | *Arribas-Hernández et al., 2021* | | |
| Genetic reagent (*A. thaliana*) | *ect2-1/ECT2pro:ECT2-mCherry-ECT2ter* | *Arribas-Hernández et al., 2018*; *Arribas-Hernández et al., 2020* | N2110839, N2110840 | Donated to NASC and ABRC |
| Genetic reagent (*A. thaliana*) | *te234/ECT2pro:ECT2$^{W464A}$-mCherry-ECT2ter* | *Arribas-Hernández et al., 2018*; *Arribas-Hernández et al., 2020* | N2110847, N2110848 | Donated to NASC and ABRC |
| Genetic reagent (*A. thaliana*) | *GFP:WIP1* | *Xu et al., 2007* | | |
| Genetic reagent (*A. thaliana*) | *ect2-1/+ GFP:WIP1ECT2pro:ECT2-mCherry-ECT2ter* | This paper (see Methods) | | Seed requests to pbrodersen@bio.ku.dk |
| Antibody | anti-mCherry (rabbit polyclonal) | Abcam | ab183628 | Used for WB (1:1000) |
| Chemical compound, drug | Glufosinate-ammonium (PESTANAL) | Sigma | 45520 77182-82-2 | Used for selection of transgenic lines |
| Sequence-based reagent | USER cloning primers | This paper (Appendix) | | Used for cloning. Sequences are in the Appendix |
| Commercial assay or kit | RNeasy Plus Micro kit (inc RLT buffer) | QIAGEN | Cat. # 74,034 | Used for RNA-extraction of FACS-sorted protoplasts |
| Commercial assay or kit | Illumina DNA Nextera Flex kit (now called Illumina DNA Prep) | Illumina | Cat. # 20018704 | Used for Smart-seq2 library preparation |
| Commercial assay or kit | NEBNext small RNA library prep set for Illumina | NEB | Cat. # E7330S | Used for sRNA-seq library preparation |
| Software, algorithm | R | https://www.R-project.org/ | | Used for data analyses |
| Software, algorithm | hyperTRIBE**R** | *Rennie et al., 2021*; https://github.com/sarah-ku/hyperTRIBER; https://github.com/sarah-ku/targets_arabidopsis | | Used for calling significant ADAR-edited sites |
| Software, algorithm | *cutadapt* | *Kechin et al., 2017* | | Used for trimming Smart-seq2, mRNAseq and sRNA-seq reads |
| Software, algorithm | STAR | *Dobin et al., 2013* | | Used for mapping Smart-seq2, mRNAseq and sRNA-seq reads |

*Continued on next page*

*Continued*

| Reagent type (species) or resource | Designation | Source or reference | Identifiers | Additional information |
|---|---|---|---|---|
| Software, algorithm | nanoPARE and PAS analysis pipelines | *Schon et al., 2018*; https://github.com/Gregor-Mendel-Institute/nanoPARE; https://github.com/maschon0/ect_polyA_analysis | | Used for PAS/PAC analyses |
| Software, algorithm | *bedtools* | *Quinlan and Hall, 2010* | | Used to merge polyadenylation clusters |
| Software, algorithm | Salmon | *Patro et al., 2017* | | Used for transcript quantification (Smart-seq2 and mRNAseq) |
| Software, algorithm | *featureCounts* | *Liao et al., 2014* | | Used for sRNA-seq quantification |
| Software, algorithm | DESeq2 | *Love et al., 2014* | | Used for DEA (Smart-seq2, mRNAseq and sRNA-seq) |
| Software, algorithm | *gprofiler2* | *Raudvere et al., 2019* | | Used for GO-term enrichment analysis |
| Software, algorithm | *ggplot2* | https://ggplot2.tidyverse.org | | Used to generate plots |
| Software, algorithm | ImageJ | *Schindelin et al., 2012* | | Used to generate fluorescence intensity plots |
| Software, algorithm | IGV (Integrative Genomics Viewer) | *Robinson et al., 2011* | | Used to navigate and represent genomic data |

All data analyses were performed using TAIR10 and Araport11 as the reference genome and transcriptome, respectively. Unless otherwise stated, data analyses were performed in R (https://www.R-project.org/) and plots generated using either base R, IGV (for genomic data) (*Robinson et al., 2011*), or *ggplot2* (https://ggplot2.tidyverse.org).

## Definitions of experiment, biological replicates, and technical replicates

We use the terms 'biological replicate' (the only type of replicate present in this study), and 'experiment' in the same way as in the accompanying manuscript (*Arribas-Hernández et al., 2021*).

## Plant material

All lines employed in this study are in the *Arabidopsis thaliana* Col-0 ecotype. The mutant alleles or their combinations: *ect2-1* (SALK_002225) (*Arribas-Hernández et al., 2018*; *Scutenaire et al., 2018*; *Wei et al., 2018*), *ect3-1* (SALK_077502), *ect4-2* (GK_241H02), and *ect2-1/ect3-1/ect4-2* (*te234*) (*Arribas-Hernández et al., 2018*) have been previously described. The transgenic lines *ect2-1 ECT2-FLAG-ADAR* and *te234 ECT2-FLAG-ADAR* (*Arribas-Hernández et al., 2021*), *GFP:WIP1* (*Xu et al., 2007*) and those expressing *ECT2pro:ECT2-mCherry-ECT2ter* and *ECT2pro:ECT2^W464A^-mCherry-ECT2ter* (*Arribas-Hernández et al., 2018*; *Arribas-Hernández et al., 2020*) have also been described. Plants co-expressing *ECT2-mCherry* and GFP-*WIP1* used for fluorescence microscopy were the F1 progeny of a genetic cross between GFP-*WIP1* and *ECT2-mCherry*-expressing plants.

## Growth conditions

Seeds for HyperTRIBE, protoplast-FACS sorting and differential expression analyses of intact root tips were sterilized, stratified, and germinated, and seedlings were grown, in identical conditions than those described for HyperTRIBE and CLIP experiments in the accompanying manuscript (*Arribas-Hernández et al., 2021*). Similarly, we also used the growth conditions described in the corresponding section of the same article for phenotypic characterization of plants expressing *ECT3-FLAG-ADAR*.

## Generation of transgenic lines for ECT3-HyperTRIBE

We generated lines expressing *ECT3pro:ECT3-FLAG-DmADAR^E488Q^cd-ECT3ter* and *ECT3pro:FLAG-DmADAR^E488Q^cd-ECT3ter* by USER cloning (*Bitinaite and Nichols, 2009*) and agrobacterium-mediated

transformation in the same way as for the ECT2 equivalents (*Arribas-Hernández et al., 2021*). Primer sequences and their combination to obtain cloning fragments are detailed in Appendix 1. As for ECT2-HT, we selected five independent lines of each type based on segregation studies of sensitivity to glufosinate-ammonium (to isolate single T-DNA insertions), phenotypic complementation (in the *te234* background), and transgene expression levels.

## HyperTRIBE

The HyperTRIBE experiments were performed once, using five biological replicates (independent lines) for each of the groups (genotypes) used. Growth conditions and experimental procedures were identical for all the groups compared in this study. Root and aerial tissue were dissected from the same plants in all cases. Tissue dissection from of 10-day-old T2 seedlings, RNA extraction, and library preparation (by Novogene) were done as described for ECT2-HyperTRIBE (*Arribas-Hernández et al., 2021*).

## Analysis of HyperTRIBE data

Significant differentially edited sites between *ECT3-FLAG-ADAR* (fusion) and *FLAG-ADAR* (control) samples were called according to our HyperTRIBE**R** pipeline (*Rennie et al., 2021*) as described for ECT2-HyperTRIBE (*Arribas-Hernández et al., 2021*), without removal of any sample. Specific scripts for this analysis can be found at https://github.com/sarah-ku/targets_arabidopsis; *Rennie, 2021*.

For the analysis of editing sites by ECT2/3-FLAG-ADAR in triple (*te234*) versus single (either *ect2-1* or *ect3-1*) mutant background, the HyperTRIBE**R** pipeline (*Rennie et al., 2021*) was run between the two types of samples without taking into account the free ADAR controls, in order to detect positions edited preferentially in one or the other background. To account for low power as a result of high variance in editing proportions due to transgene expression differences across samples, scaled ADAR abundance was treated as an extra covariate in the model. This resulted in enriched sensitivity to specifically call A-to-G positions, which were subsequently considered as significant if they had an adjusted p-value<0.1 and an absolute $\log_2$(fold change)>0.25. We also required positions to be a significantly edited site in at least one of the single or triple mutant set ups against the free FLAG-ADAR control samples.

## Comparison with root single-cell data

The expression matrix based on a total of 4727 individual cells from scRNA-seq in roots was downloaded from *Denyer et al., 2019*, together with extensive lists of marker genes associated with 15 clusters annotated to cell types in roots. To calculate the proportion of markers at target genes: for each of the 15 clusters, the proportion of marker genes that are ECT2 or ECT3 targets (based on ECT2-HT and ECT3-HT, respectively, in roots) was calculated. Proportions were then overlayed onto a t-SNE diagram (*Denyer et al., 2019*), according to relevant clusters of cells. Scripts for this analysis can be found at https://github.com/sarah-ku/targets_arabidopsis.

## Preparation and sorting of protoplasts

We harvested roots from 5-day-old T4 seedlings grown on vertical square plates (20 plates with 4 rows of densely spotted seeds in each plate per line/replicate) to digest in 20 ml of protoplasting solution (20 mM MES, 0.4 M D-Mannitol, 20 mM KCl, 1.25% w/v Cellulase, 0.3 % Macerozyme, 0.1% w/v BSA, 10 mM CaCl2, and 5 mM β-mercaptoethanol; pH 5.7), following Benfey's lab procedure (*Birnbaum et al., 2005*; *Bargmann and Birnbaum, 2010*). After 75 min of incubation at 27 °C with gentle agitation, we filtered the cell-suspensions through a 40 μm strainer and pelleted cells by centrifugation at 500×*g* for 10 min at room temperature in a swinging-bucket centrifuge. Pellets were gently resuspended in 400 μl of protoplasting solution for direct sorting in a FACSArialII cytometer. The flow stream was adjusted to 20 psi sheath pressure with a 100 μm nozzle aperture. Sorted cells were collected into RTL buffer (QIAGEN) supplemented with 40 mM DTT (3.5 vol of buffer per volume of cell suspension) and lysed by vortexing. The protoplast extracts were flash-frozen on dry ice until extraction with the RNeasy Plus Micro Kit (QIAGEN) following the manufacturer's instructions. The yield was ~300,000 cells in a volume of 1.5 ml (per sample). Samples were harvested, prepared, and sorted with a 15 min lapse between them to account for sorting time. In that way, every sample was

processed in the same amount of time (~2 h from the start of harvesting to sorting). To prevent any possible bias, the samples of each genotype (3+3) were alternated during all the processing.

## Smart-seq2

Smart-seq2 libraries were generated according to *Picelli et al., 2013* using the Illumina DNA Nextera Flex Kit (now called Illumina DNA Prep) from total RNA extracted with the RNeasy Plus Micro Kit (QIAGEN) from FACS-sorted root protoplasts (*Birnbaum et al., 2005*). The libraries were sequenced in PE75 mode on an Illumina NextSeq550 sequencer. Nextera transposase adapters were trimmed from all reads using *cutadapt* (*Kechin et al., 2017*).

## Polyadenylation site analysis

Smart-seq2 reads with at least nine 3'-terminal A nucleotides or 5'-terminal T nucleotides were labeled as putative poly(A)-containing reads and the oligo-A/T sequences were removed with a maximum allowed mismatch rate of 6 %. All putative poly(A)-containing reads with a length >20 and a mean quality score >25 after trimming were retained along with their mate pair and mapped to the *A. thaliana* TAIR10 genome using STAR (*Dobin et al., 2013*) with the following parameters:

```
--alignIntronMax 5000 --alignMatesGapMax 5500 --outFilterMatchNmin 20
--alignSJDBoverhangMin 1 --outFilterMismatchNmax 5
--outFilterMismatchNoverLmax .05 --outFilterType BySJout
--outFilterIntronMotifs RemoveNoncanonicalUnannotated
```

Putative poly(A)-containing reads that mapped to the genome were filtered for false positives by examining the adjacent nucleotides in the genome: reads were removed if the putative poly(A) site was immediately upstream of a 15 nt region that is at least 80 % purines (which are likely sites of oligo-dT mispriming). All putative poly(A)-containing reads not filtered in this way were retained as poly(A) sites and were counted for each position in the genome based on the most 3' nucleotide of each read (allowing 3'-terminal mismatches).

Polyadenylation site clusters (PACs) were identified using a modification of the nanoPARE analysis pipeline (*Schon et al., 2018*) (specific scripts for this analysis are available at https://github.com/maschon0/ect_polyA_analysis; *Schon, 2021*). Briefly, reads from the samples above that did not contain untemplated poly(A) tails were mapped to the genome and used as a negative control of 'gene body reads.' Then, subtractive kernel density estimation was performed for each sample using *endGraph.sh* with default settings to produce a BED file for each sample containing poly(A) site clusters. As a final filter against oligo-dT mispriming events, the reads removed as false positives in the previous step that overlap with each cluster were counted. If the cluster contained more filtered signal than unfiltered signal, the entire cluster was considered a false positive cluster. Clusters were retained if an overlapping site was identified in at least two of the three replicates of both *ect2-1/ECT2-mCherry* and *te234/ECT2*[W464A]*-mCherry* genotypes. These two sets of clusters were merged using *bedtools merge* (*Quinlan and Hall, 2010*). Clusters mapping to the mitochondrial and chloroplast genome and the two rDNA loci were discarded, and the rest were retained for quantification with Salmon (*Patro et al., 2017*).

## mRNA-seq and small RNA-seq from root tips

Total RNA purified from manually dissected root tips of 4-day-old plants (using the same growing conditions and methodology as for the HyperTRIBE lines) was used for the preparation of Illumina mRNA-seq (same methodology as for HyperTRIBE) and small RNA-seq libraries (NEBNext small RNA library prep set for Illumina). The experiment was performed once, using three biological replicates for mRNA-seq, and two for small RNA-seq.

## Differential expression analysis (mRNA and miRNAs)

Differential gene expression analysis of mRNA was performed from processed (*cutadapt Kechin et al., 2017*), mapped (STAR, *Dobin et al., 2013*) and quantified (Salmon, *Patro et al., 2017*) Smart-seq2 or mRNA-seq data using DESeq2 (*Love et al., 2014*), for all genes with at least 1 TPM in all six samples (three biological replicates of the two types) and a total sum of at least 5 TPM. Significantly differentially expressed genes (FDR<0.05) were considered to be upregulated in the mutants if the fold change between mutant and wild type samples was higher than 1.5, or downregulated if lower than 1/1.5.

For small RNA-seq, raw reads were trimmed with *cutadapt* (*Kechin et al., 2017*) to lengths of 18–28 nt, and mapped to the *Arabidopsis* genome using STAR (*Dobin et al., 2013*) with genome indexes built on the Araport11_GTF_genes_transposons.Mar202021.gtf annotation. Mapped reads were counted using *featureCounts* (*Liao et al., 2014*). Genes with less than 1 RPM in all four samples (two biological replicates of the two types) were excluded from the analysis. Differential expression analysis was conducted using DESeq2 (*Love et al., 2014*) on the resulting matrix. Genes with adjusted p-value (FDR) lower or equal to 0.05 were considered significantly differentially expressed, and upregulated in *te234* mutants if the fold change between *te234* and *Col-0 WT* samples was higher than 1.5, or downregulated if lower than 1/1.5.

## Gene ontology enrichment analysis

The functional enrichment analysis was carried out using the R package *gprofiler2* (*Raudvere et al., 2019*).

## Fluorescence microscopy

Entire root tips growing inside MS-agar plates were imaged with a Leica MZ16 F stereomicroscope equipped with a Sony α6000 camera. Standard confocal fluorescence microscopy images of cells in root meristems were acquired with a Zeiss LSM700 confocal microscope as described in *Arribas-Hernández et al., 2018* using ~7-day-old seedlings grown on MS-agar plates and freshly mounted in water. For super-resolution fluorescence microscopy, we used a Zeiss LSM900 equipped with the Airyscan detector (*Huff, 2015*). Fluorescence intensity plots were obtained with the tool 'Plot Profile' of the image-processing package ImageJ (*Schindelin et al., 2012*).

## Acknowledgements

The authors thank Lena Bjørn Johansson and Phillip Andersen for technical assistance in the construction of transgenic lines, Emilie Oksbjerg for preparing and sequencing small RNA libraries, Theo Bølsterli, René Hvidberg Petersen and their teams for plant care, and Anna Fossum and Rajesh Somasundaram for their assistance with FACS. Kim Rewitz and Kenneth Halberg are thanked for their assistance with Airyscan microscopy. The authors acknowledge Mathias Tankmar, Alexander JH Andersen, and Freja Asmussen for experimental support, and Tom Denyer and Marja Timmermans for their input and support in the analyses of ECT2/3 expression and target enrichment in their scRNAseq data. The authors are grateful to Norman R Groves and Iris Meier for the kind donation of GFP-WIP1 seeds.

## Additional information

### Funding

| Funder | Grant reference number | Author |
|---|---|---|
| H2020 European Research Council | PATHORISC ERC-2016-COG 726417 | Peter Brodersen |
| Independent Research Fund Denmark | 9040-00409B | Peter Brodersen |
| H2020 European Research Council | 638173 | Robin Andersson |
| Independent Research Fund Denmark | 6108-00038B | Robin Andersson |
| H2020 European Research Council | 63788 | Michael D Nodine |

The funders had no role in study design, data collection and interpretation, or the decision to submit the work for publication.

## Author contributions
Laura Arribas-Hernández, Conceptualization, Investigation, Methodology, Project administration, Supervision, Validation, Visualization, Writing – original draft, Writing – review and editing; Sarah Rennie, Data curation, Formal analysis, Investigation, Methodology, Software, Supervision, Writing – original draft, Writing – review and editing; Michael Schon, Data curation, Investigation, Methodology, Software, Writing – original draft; Carlotta Porcelli, Data curation, Formal analysis, Investigation, Methodology, Writing – review and editing; Balaji Enugutti, Methodology; Robin Andersson, Funding acquisition, Supervision, Writing – review and editing; Michael D Nodine, Conceptualization, Funding acquisition, Supervision, Writing – review and editing; Peter Brodersen, Conceptualization, Funding acquisition, Investigation, Methodology, Project administration, Supervision, Visualization, Writing – original draft, Writing – review and editing

## Author ORCIDs
Laura Arribas-Hernández http://orcid.org/0000-0003-0605-0407
Carlotta Porcelli http://orcid.org/0000-0003-4675-4898
Balaji Enugutti http://orcid.org/0000-0002-0816-024X
Robin Andersson http://orcid.org/0000-0003-1516-879X
Michael D Nodine http://orcid.org/0000-0002-6204-8857
Peter Brodersen http://orcid.org/0000-0003-1083-1150

## Decision letter and Author response
Decision letter https://doi.org/10.7554/eLife.72377.sa1
Author response https://doi.org/10.7554/eLife.72377.sa2

---

# Additional files

## Supplementary files
- Supplementary file 1. ECT3 HyperTRIBE data.
- Supplementary file 2. ECT2 and ECT3 HyperTRIBE data in triple versus single mutants.
- Supplementary file 3. ECT2/ECT3-target sets.
- Supplementary file 4. Poly(A) site selection data.
- Supplementary file 5. Differential expression analysis of *ect2/ect3/ect4* mutants (mRNA).
- Supplementary file 6. Differential expression analysis of *ect2/ect3/ect4* mutants (miRNAs).
- Transparent reporting form

## Data availability
Accession numbers: The raw and processed data for ECT3-HyperTRIBE, Smart-seq2 from root protoplasts and RNA-seq from root tips have been deposited in the European Nucleotide Archive (ENA) at EMBL-EBI under the accession number PRJEB44359.
Code availability: The scripts for the HyperTRIBE and scRNA-seq analyses can be found at https://github.com/sarah-ku/targets_arabidopsis (copy archived at https://archive.softwareheritage.org/swh:1:rev:ad524fd57073b569320998bb79ecc66d433b37c7), and for the PAS analysis at https://github.com/maschon0/ect_polyA_analysis (copy archived at https://archive.softwareheritage.org/swh:1:rev:f469a02ac12ed83f442c9936b46c4f94336c10c1).

The following dataset was generated:

| Author(s) | Year | Dataset title | Dataset URL | Database and Identifier |
|---|---|---|---|---|
| Brodersen P | 2021 | Principles of mRNA targeting and regulation via Arabidopsis YTHDF proteins | https://www.ebi.ac.uk/ena/browser/view/PRJEB44359 | European Nucleotide Archive, PRJEB44359 |

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

## Appendix 1

**Cloning**

| Constructs | Primer pairs (fragments) |
|---|---|
| *ECT3pro:ECT3-FLAG-ADAR-ECT3ter* (in pCAMBIA3300-U) | LA617-692 (*ECT3pro:ECT3*), LA693-642 (*FLAG-ADAR*), and LA626-622 (*ECT3ter*) |
| *ECT3pro:FLAG-ADAR-ECT3ter* (in pCAMBIA3300-U) | LA617-627 (*ECT3pro*), LA694-642 (*FLAG-ADAR*), and LA626-622 (*ECT3ter*) |

| Primers for USER-cloning | |
|---|---|
| LA617.U-ECT3P.F | GGCTTAAUAGGCTTGGTTAGCAGAAGG |
| LA622.ECT3T-U.R | GGTTTAAUTGGTTAACTCTATGGACTCATC |
| LA626.ADAR/ECT3T.F | AACCAACCUTTGGTTTTAAGTGGGAAC |
| LA627.ECT3P/FLAG.R | ATGGCTGCGUAGTGAGTGGCTTAG |
| LA642.dADAR/ECT3T.R | AGGTTGGTUCATTCGGCAAGACCGAACTC |
| LA692.ECT3/FLAG.R | AGGCAGUAGCAACAGCATTTTTCTCC |
| LA693.ECT3/FLAG.F | ACTGCCUGCGATTACAAGGATGACGATGAC |
| LA694.ECT3P/FLAG.F | ACGCAGCCAUGGATTACAAGGATGACGATG |

