## [Decision Letter]

**Acceptance summary:**

In this work, you addressed contradictory reports regarding the subcellular location of the YTHDF proteins ECT2 and ECT3 proteins. The work also identified the target of these proteins, scored the redundancy among them and their biological functions as m6A-RNA binding proteins. We found your work of general interest with an important contribution to the fields of plant and RNA biology.

**Decision letter after peer review:**

Thank you for submitting your article "The Arabidopsis m^6^A-binding proteins ECT2 and ECT3 bind largely overlapping mRNA target sets and influence target mRNA abundance, not alternative polyadenylation" for consideration by *eLife*. Your article has been reviewed by 2 peer reviewers, including Pablo A Manavella as Reviewing Editor and Reviewer #1, and the evaluation has been overseen by Jürgen Kleine-Vehn as the Senior Editor. The following individual involved in review of your submission has agreed to reveal their identity: Rupert Fray (Reviewer #2).

Essential revisions:

Similar to the companion manuscript both reviewers considered this a valuable and well-conducted paper. For this article we focused the discussion in whether it was better to keep both manuscript as independent articles or combine them into a single paper similar to your original BioRxiv pre-print. After discussing this point we reached the conclusion that a combined article would be hard to digest by the readers and many important data may end up diluted in such an extended article. Thus we decided to consider both submissions individually. However, there are some issues we will like you to tackle before resubmission.

1. We have noticed that some sessions and concepts across this manuscript are rather repetitive with the co-submitted manuscript. Considering that it is likely that both manuscripts end up accepted for publication back-to-back, we recommend you reshaping the manuscripts to avoid repetitive content and instead direct the reader to the other paper.

2. Regarding the data in Figure 8 (miRNA profile in root tips of ect2/ect3/ect4 knockout plants). We wondered whether the authors found ETC2/3 bound to pri-miRNAs in the datasets? even when most pri-miRNAs are processed in the nucleus, there are reports of cytoplasmic pri-miRNAs (for example those encoding miR-PEP) and recently m6A was reported to affect the miRNA pathway (doi: 10.1073/pnas.2003733117). Perhaps this can be further explored/discussed.

3. It could also be useful in the introduction and discussion to elaborate on the potential functions of the other eight members of the YTHDF family. The authors have made a convincing case that the ECT2 to 4 have redundant functions, do they know if ECTs from the other groups are able to complement the triple mutant if driven by a constitutive or ECT2 promoter? We don't think additional experiments are required here, but a reader could come away with the feeling that all YTHDF types are likely carrying out the same function in Arabidopsis, which is not necessarily clear yet.

---

## [Author Response]

Essential revisions:Similar to the companion manuscript both reviewers considered this a valuable and well-conducted paper. For this article we focused the discussion in whether it was better to keep both manuscript as independent articles or combine them into a single paper similar to your original BioRxiv pre-print. After discussing this point we reached the conclusion that a combined article would be hard to digest by the readers and many important data may end up diluted in such an extended article. Thus we decided to consider both submissions individually. However, there are some issues we will like you to tackle before resubmission.

We are happy that you agree with our decision to split reporting of our results on ECT2 and ECT3 through transcriptomic methods into two separate manuscripts. As you are aware, we initially attempted to report everything in one paper, but found that it lost focus as a consequence of trying to achieve too much in one single report. Several colleagues told us the same after reading our original pre-print from April 2021 on BioRxiv. We would like to take the opportunity to stress that our decision to split was only motivated by our wish to communicate our findings and conclusions in the clearest possible way, not by a wish to inflate our CVs.

1. We have noticed that some sessions and concepts across this manuscript are rather repetitive with the co-submitted manuscript. Considering that it is likely that both manuscripts end up accepted for publication back-to-back, we recommend you reshaping the manuscripts to avoid repetitive content and instead direct the reader to the other paper.

Since the same (reasonable) point was made on the companion manuscript, we insert the answer to that point below:

We have done our very best avoid repetition between the two manuscripts. In particular, we have removed all non-essential information in the Introduction of each paper that is already presented in the other, and eliminated redundancy in the Methods of the companion manuscript referring to the equivalent information in this one (that has been unified as a unique Methods section in the main text to eliminate the original Supplemental Methods file). Avoiding repetitive content in the Results section has been more problematic though: ECT3 HyperTRIBE results are only shown and analyzed in this paper, but since it is an independent data set, it must be subjected to the same rigorous controls as is the ECT2 data set described in the companion paper. Because the results for both proteins are similar, that inevitably causes a similar structure of those parts of the papers that describes the HyperTRIBE data – we do not see a way around that. Nonetheless, we have shortened the description of ECT3 HyperTRIBE in this paper by referring to the corresponding parts of the companion paper whenever possible.

2. Regarding the data in Figure 8 (miRNA profile in root tips of ect2/ect3/ect4 knockout plants). We wondered whether the authors found ETC2/3 bound to pri-miRNAs in the datasets? even when most pri-miRNAs are processed in the nucleus, there are reports of cytoplasmic pri-miRNAs (for example those encoding miR-PEP) and recently m6A was reported to affect the miRNA pathway (doi: 10.1073/pnas.2003733117). Perhaps this can be further explored/discussed.

We thank the reviewers for raising this point. We agree that the possibility of direct ECT2/3 binding to pri-miRNAs corresponding to the miRNAs whose expressing is altered in ect2/3/4 mutants should have been examined and commented on, particularly given recent reports on m6A in certain pri-miRNA transcripts. We have now re-examined our iCLIP and HyperTRIBE data for evidence of ECT2/3 binding to primiRNAs.

We see no such evidence, and have included a comment to state as much in the section of Results describing the miRNA expression profiles. We also note that because ECT2/3 HyperTRIBE was conducted in plants with a functional miRNA biogenesis pathway in which pri-miRNAs turn over rapidly by dicing, the failure to detect ECT2/3:pri-miRNA association should not be taken as rigorous proof that such association does not take place.

3. It could also be useful in the introduction and discussion to elaborate on the potential functions of the other eight members of the YTHDF family. The authors have made a convincing case that the ECT2 to 4 have redundant functions, do they know if ECTs from the other groups are able to complement the triple mutant if driven by a constitutive or ECT2 promoter? We don't think additional experiments are required here, but a reader could come away with the feeling that all YTHDF types are likely carrying out the same function in Arabidopsis, which is not necessarily clear yet.

We agree that one should, in general, be very careful when concluding on redundancy in vivo. We also agree that the case for genuine redundancy between ECT2 and ECT3 in promotion of cellular proliferation made here neither means that ECT2 and ECT3 always act redundantly, nor does it mean that the idea of redundant YTHDF function extends to all ECT paralogs in Arabidopsis. To clarify this, we have extended the discussion of this point, including among other a consideration of the fact that specific ECT paralogs are expressed in cells with very different properties, perhaps indicating distinct function, and nearly certainly indicating distinct target sets. For example, while ECT2/3 are expressed in rapidly proliferating cells of organ primordia, ECT10 is the sole ECT protein highly expressed in pollen.